# Riemannian Dueling Optimization

**Yuxuan Ren** [1]  **Abhishek Roy** [2]  **Shiqian Ma** [1]

## Abstract

Dueling optimization considers optimizing an objective with access to only a comparison oracle of the objective function. It finds important applications in emerging fields such as recommendation systems and robotics. Existing works on dueling optimization mainly focused on unconstrained problems in the Euclidean space. In this work, we study dueling optimization over Riemannian manifolds, which covers important applications that cannot be solved by existing dueling optimization algorithms. In particular, we propose a Riemannian Dueling Normalized Gradient Descent (RDNGD) method and establish its iteration complexity when the objective function is geodesically $L$-smooth or geodesically (strongly) convex. We also propose a projection-free algorithm, named Riemannian Dueling Frank–Wolfe (RDFW) method, to deal with the situation where projection is prohibited. We establish the iteration and oracle complexities for RDFW. We illustrate the effectiveness of the proposed algorithms through numerical experiments on both synthetic and real applications.

## 1. Introduction

Many modern learning tasks involve settings where gradients or even function values are inaccessible, and the only feasible feedback comes in the form of pairwise preferences or comparison – often referred to as *dueling feedback*. For example, users in recommendation systems typically express relative judgments such as "item A is preferred over item B"; in robotics, human supervisors often compare trajectories rather than assign scalar rewards (Yang et al., 2025; Jain et al., 2013); and in representation learning, a downstream classifier may indicate that one projection

preserves class structure better than another (Ventocilla & Riveiro, 2020; Morariu et al., 2021). Importantly, in many of these applications the decision space itself is inherently non-Euclidean: recommendation models often rely on hierarchical embeddings in hyperbolic space (Chamberlain et al., 2019; Shimizu et al., 2024), trajectory optimization in robotics often requires optimizing over special orthogonal groups SO(3) (Watterson et al., 2018), and projection matrices for representation learning are naturally constrained to the Stiefel manifold (Boumal, 2023). In addition, solutions to high-dimensional learning problems frequently lie on low-dimensional manifolds (Garipov et al., 2018; Hu et al., 2022). Furthermore, several preference elicitation frameworks require constrained optimization where feasible solutions belong to constraint sets such as sphere, and simplex. These problems can also be reformulated as optimization over a manifold (Boumal, 2023). Classical problems such as the Karcher mean problem where the feedback is provided by human and they are not aware of the loss can also be formulated as a Riemannian dueling optimization problem. Motivated by these applications, we study the Riemannian dueling optimization problem in the following form:

$$\min_{x \in \mathcal{X} \subseteq \mathcal{M}} f(x), \tag{1}$$

where $\mathcal{M}$ is a Riemannian manifold with dimension $d$, and $f : \mathcal{X} \subseteq \mathcal{M} \to \mathbb{R} \cup \{\infty\}$ is a smooth function. Let $x^*$ denote the minimizer of (1), and $f^* := f(x^*)$. We assume that we have access to only a pairwise comparison oracle $\mathcal{Q}_f(x, y)$ between two queried points $x, y \in \mathcal{X} \subseteq \mathcal{M}$ given by

$$\mathcal{Q}_f(x, y) = 2 * \mathbb{1}(f(x) > f(y)) - 1, \tag{2}$$

where $\mathbb{1}(\cdot)$ is the indicator function, which equals 1 if the statement is true and 0 otherwise. Note that we do not have access to the function values $f(x)$ and $f(y)$.

### 1.1. Review of Euclidean Dueling Optimization

In this subsection, we briefly review the literature on Euclidean dueling optimization. (Lobanov et al., 2024) and (Chervonenkis et al., 2024) proposed coordinate descent algorithms for dueling optimization. (Bergou et al., 2020) introduced a stochastic three points method which is a direct search algorithm based on comparison oracles. Variants of

[1]Department of Computational Applied Mathematics and Operations Research, Rice University, Houston, USA [2]Department of Statistics, Texas A&M University, College Station, USA. Correspondence to: Yuxuan Ren <yuxuan.ren@rice.edu>.

*Proceedings of the 43rd International Conference on Machine Learning*, Seoul, South Korea. PMLR 306, 2026. Copyright 2026 by the author(s).

this method were further studied in (Boucherouite et al., 2024; Kadi & Saadi, 2025; Tkachenko et al., 2024). These methods typically rely on identifying the best candidate among the current iterate and its neighbors, which requires multiple calls to the comparison oracle in each iteration. The main difficulty in dueling optimization is how to efficiently estimate the gradient direction using comparison oracle, and recent advances have explored sophisticated techniques for explicit gradient approximation. (Cai et al., 2022) proposed one-bit and comparison-based gradient estimators; (Tang et al., 2024) proposed to estimate gradient using ranking oracles; (Tao et al., 2026) proposed a binary search method to compute the gradient estimator using comparison oracle. In our work, we extend the two-point random perturbation approach for gradient estimation proposed by Saha et al. (2021) to the Riemannian setting, as it requires only a single random direction and one comparison per iteration. However, extending this framework is far from a trivial generalization, as the presence of curvature breaks standard Euclidean trigonometry and linearization.

## 1.2. Motivating Examples

We now discuss two motivating examples for Riemannian dueling optimization.

**Attack on Deep Neural Network.** Attacks on deep neural networks (DNNs) aim to introduce small, human-imperceptible perturbations to an input so that the model produces an incorrect label. In realistic black-box scenarios, the attacker typically has only limited query access to the model and therefore cannot rely on gradient information. Moreover, the attacker may not even have access to a clean or reliable loss function. For example, prediction APIs often return only labels or heavily processed scores (Xie et al., 2018; Dhillon et al., 2018; Athalye et al., 2018), so function values are either unavailable or not reliable. In such cases, it is more natural to ask which of two perturbed images is more adversarial, and to use these pairwise preferences to drive the optimization. Furthermore, natural images are widely believed to concentrate near low-dimensional manifolds (Weinberger & Saul, 2006). Similar to the optimal $\ell_2$-norm attack framework established in (Lyu et al., 2015) and further explored in (Li et al., 2023), we model the constraint manifold $\mathcal{M}$ as a sphere for simplicity. This leads to the following dueling optimization in the form of (1):

$$\max_{x \in \mathcal{M}} \quad f(x) := \mathcal{L}_w(x_0 + x),$$

where $\mathcal{L}_w$ denotes the loss function of a DNN with weights $w$, $x_0$ is the image to attack, and the decision variable $x$ represents the adversarial perturbation.

**Horizon Leveling.** Horizon leveling is a fundamental task in computer vision, where the goal is to estimate and correct the image horizon. It plays a key role in applications such as single-image camera calibration (Hold-Geoffroy et al., 2018) and visual navigation (Ettinger et al., 2002). Formally, the horizon tilt in an image can be represented by a rotation matrix $R_{\text{tilt}} \in \text{SO}(2)$, where a perfectly level horizon corresponds to the identity matrix $I$. In many realistic settings, however, an explicit loss function or ground-truth labels may be unavailable. Instead, we observe only pairwise comparisons from the loss function $f(R)$ indicating which of two rotated versions appears more level. This leads naturally to the following Riemannian dueling optimization problem over $\text{SO}(2)$, where the goal is to find the correction $R^\star$ that best compensates for the observed tilt $R_{\text{tilt}}$.

$$\min_{R \in \text{SO}(2)} f(R).$$

*Remark* 1.1. Our applications illustrate distinct feedback sources: machine-generated adversariality (DNN attacks) versus human preferences (horizon leveling). This demonstrates that our framework is agnostic to the comparison source, unifying both model-based and human-based objectives within a single scheme.

## 1.3. Contributions

In this paper, we introduce Riemannian dueling optimization, where only a comparison oracle of the objective function is available, and thus classical Riemannian optimization algorithms fail. Our main contributions are as follows.

- We propose the first Riemannian dueling normalized gradient descent method (RDNGD) for Riemannian dueling optimization and provide its iteration and oracle complexities for geodesically $L$-smooth objectives, as well as for geodesically $L$-smooth and (strongly) geodesically convex objectives.

- We propose a Riemannian dueling Frank-Wolfe method (RDFW), which is projection-free, to handle cases where projection is computationally prohibitive. We establish iteration and oracle complexities of RDFW in the geodesically convex setting, which is the first convergence result for projection-free dueling optimization over manifolds.

Overall, this work bridges two active areas, preference-based optimization and Riemannian optimization, by showing how optimization can proceed reliably on manifolds when only comparison oracles are available. Our main results are summarized in Table 1. Our RDNGD and RRDNGD algorithms can be viewed as Riemannian generalizations of the NGD methods proposed in (Saha et al., 2021). While we employ a similar gradient direction estimator, our contribution extends significantly beyond a trivial generalization to the Riemannian setting. In fact, when reducing to the Euclidean setting, we also provide better results comparing with the results in (Saha et al., 2021), and we will specify them when we present the specific results.

*Table 1.* Comparison of the proposed algorithms with existing methods.

| Algorithm | Requirements on $f$ | Requirements on $\mathcal{X}$ | Constraint Handling | Iteration Complexity | Oracle Complexity |
|---|---|---|---|---|---|
| **RDNGD (nonconvex)** | g-$L$-smooth | Unconstrained ($\mathcal{X} = \mathcal{M}$) | / | $\mathcal{O}(d\epsilon^{-2})$ | $\mathcal{O}(d\epsilon^{-2})$ |
| **RDNGD (convex)** | g-$L$-smooth, g-convex | Geodesically uniquely convex | Projection | $\mathcal{O}(d\epsilon^{-1})$ | $\mathcal{O}(d\epsilon^{-1})$ |
| **RRDNGD** | g-$L$-smooth, strongly g-convex | Geodesically uniquely convex | Projection | $\mathcal{O}(d\log(1/\epsilon))$ | $\mathcal{O}(d\log(1/\epsilon))$ |
| **RDFW** | g-$L$-smooth, g-convex | Geodesically uniquely convex | LMO | $\mathcal{O}(\epsilon^{-1})$ | $\mathcal{O}(d\epsilon^{-2})$ |
| $\beta$-**NGD** (Saha et al., 2021) | $L$-smooth, convex | Unconstrained | / | $\mathcal{O}(d\epsilon^{-1})$ | $\mathcal{O}(d\epsilon^{-1})$ |
| $(\alpha, \beta)$-**NGD** (Saha et al., 2021) | $L$-smooth, strongly convex | Unconstrained | / | $\mathcal{O}(d\log(1/\epsilon))$ | $\mathcal{O}(d\log(1/\epsilon))$ |

## 2. Preliminaries

We consider smooth Riemannian manifold $\mathcal{M}$ equipped with a Riemannian metric $g$. For any point $x \in \mathcal{M}$, the metric induces an inner product on the tangent space $\mathrm{T}_x\mathcal{M}$, denoted as $\langle \cdot, \cdot \rangle_x$, and an associated norm $\|v\|_x := \sqrt{\langle v, v \rangle_x}$. The Riemannian gradient $\mathrm{grad} f(x) \in \mathrm{T}_x\mathcal{M}$ is uniquely defined as the tangent vector satisfying $\langle \mathrm{grad} f(x), v \rangle_x = \mathrm{D} f(x)[v]$ for all $v \in \mathrm{T}_x\mathcal{M}$, where $\mathrm{D} f(x)[\cdot]$ denotes the differential. To generalize the concept of straight lines in Euclidean space to the manifold, we use geodesics defined as curves that locally minimize distance between two points. The exponential map, $\mathrm{Exp}_x : \mathrm{T}_x\mathcal{M} \to \mathcal{M}$, maps a tangent vector $v$ to $y = \mathrm{Exp}_x(v)$ by following the geodesic starting at $x$ with velocity $v$ for unit time. Conversely, the logarithmic map $\mathrm{Log}_x : \mathcal{M} \to \mathrm{T}_x\mathcal{M}$ is defined as the local inverse of the exponential map, mapping a point $y$ on the manifold back to the tangent vector $v = \mathrm{Log}_x(y)$ such that the geodesic distance between $x$ and $y$ is given by $\|v\|_x$. We use $\Gamma_y^x$ to denote the parallel transport operator that moves a vector along a geodesic while preserving its norm and direction from $\mathrm{T}_y\mathcal{M}$ to $\mathrm{T}_x\mathcal{M}$. We refer the reader to Appendix A for formal definitions and detailed properties of these concepts.

Now we present some definitions and assumptions.

**Definition 2.1** (Geodesic $L$-smoothness). A differentiable function $f : \mathcal{M} \to \mathbb{R}$ is said to be geodesically $L$-smooth if $\mathrm{grad} f(x)$ is $L$-Lipschitz, i.e. for any $x, y \in \mathcal{M}$,

$$\|\mathrm{grad} f(x) - \Gamma_y^x \mathrm{grad} f(y)\|_x \leq L d(x, y).$$

This also implies,

$$|f(y) - f(x) - \langle \mathrm{grad} f(x), \mathrm{Log}_x(y) \rangle_x| \leq \frac{L}{2} d^2(x, y).$$

**Definition 2.2** (Geodesically uniquely convex set). We call $\mathcal{X} \subseteq \mathcal{M}$ a geodesically uniquely convex set if for any $x, y \in \mathcal{X}$, there exists a unique geodesic $\gamma$ such that

$$\gamma(0) = x, \quad \gamma(1) = y, \quad \text{and } \gamma(t) \in \mathcal{X} \text{ for all } t \in [0, 1].$$

**Definition 2.3** (Geodesic (strong) convexity). Let $\mathcal{X}$ be a geodesically uniquely convex set. The function $f : \mathcal{X} \subseteq \mathcal{M} \to \mathbb{R}$ is geodesically $\alpha$-strongly convex, i.e., for any $x, y \in \mathcal{X}$, it holds that

$$f(y) \geq f(x) + \langle \mathrm{grad} f(x), \mathrm{Log}_x(y) \rangle_x + \frac{\alpha}{2} d^2(x, y). \quad (3)$$

If $f$ satisfies (3) with $\alpha = 0$, then $f$ is geodesically convex.

*Remark* 2.4. We adopt the notion of the geodesically uniquely convex set $\mathcal{X}$ (Kim & Yang, 2022) to address the ambiguity associated with the possible non-uniqueness of geodesics. This relaxes the global property to a local property, enabling a rigorous definition of convexity. This also ensures that $\mathrm{Exp}_x$ is a diffeomorphism for any $x$ in the geodesically uniquely convex set, and thus ensures that $\mathrm{Log}_x(y) = \mathrm{Exp}_x^{-1}(y)$ is well-defined.

*Assumption* 2.5. The sectional curvature of $\mathcal{M}$ is bounded below by $\kappa$.

*Assumption* 2.6 (Nonexpansive projection). We assume that the set $\mathcal{X}$ is geodesically uniquely convex, and the projection oracle $\mathcal{P}_\mathcal{X} : \mathcal{M} \to \mathcal{X}$ defined as $\mathcal{P}_\mathcal{X}(x) := \{y \in \mathcal{X} : d(x, y) = \inf_{z \in \mathcal{X}} d(x, z)\}$, is nonexpansive, i.e., $d(\mathcal{P}_\mathcal{X}(x), \mathcal{P}_\mathcal{X}(y)) \leq d(x, y)$ for all $x, y \in \mathcal{M}$.

## 3. Riemannian Dueling Normalized Gradient Descent Method

In this section, we present our Riemannian Dueling Normalized Gradient Descent (RDNGD) method for solving (1). For ease of notation, we use $\langle \cdot, \cdot \rangle$ to denote the inner product $\langle \cdot, \cdot \rangle_x$ on $\mathrm{T}_x\mathcal{M}$ and $\| \cdot \|$ to denote $\| \cdot \|_x$, when there is no ambiguity. For a tangent space $\mathrm{T}_x\mathcal{M}$ with an orthonormal basis $\{e_1, e_2, \ldots e_d\}$, any $v \in \mathrm{T}_x\mathcal{M}$ can be expressed as $v = \sum_{i=1}^d v_i e_i$ for some $v_1, v_2, \ldots, v_d \in \mathbb{R}$. When the choice of basis is clear from context, we represent the tangent vector by its coefficient vector $v := [v_1, v_2, \ldots, v_d]^\top$. For a given point $x \in \mathcal{M}$, we use $\mathcal{S}_{\mathrm{T}_x\mathcal{M}}(r)$ to denote the sphere with center $x$ and radius $r$ on $\mathrm{T}_x\mathcal{M}$. For a set $S$, let $\mathrm{Unif}(S)$ represent the uniform distribution on $S$. Sam-

pling a tangent vector uniformly from $\mathcal{S}_{\mathrm{T}_x\mathcal{M}}(1)$ is equivalent with sampling the coefficient vector $u$ uniformly from $\mathcal{S}_d(1) := \{x \in \mathbb{R}^d : \|x\|_2 = 1\}$.

We now introduce our Riemannian gradient direction estimator using dueling oracles $\mathcal{Q}_f(x, y)$. Although such estimators have been proposed for the Euclidean space (Tao et al., 2026; Saha et al., 2021), none of them applies to the Riemannian setting directly. Our Riemannian gradient direction estimator is defined as:

$$h_\nu(x) = \mathcal{Q}_f(\mathrm{Exp}_x(\nu u), \mathrm{Exp}_x(-\nu u))u, \qquad (4)$$

where $\nu > 0$ is the perturbation radius and $u \sim \mathrm{Unif}(\mathcal{S}_{\mathrm{T}_x\mathcal{M}}(1))$. The exponential map ensures that the perturbed points stay on $\mathcal{M}$. $u \sim \mathrm{Unif}(\mathcal{S}_{\mathrm{T}_x\mathcal{M}}(1))$ is sampled by projecting a random Gaussian vector onto the tangent space followed by normalization. The following results establish the relation between $h_\nu(x)$ in (4) and the normalized gradient. In Lemma 3.1 we first show that $h_\nu(x)$ coincides with $\mathrm{sign}(\langle \mathrm{grad}f(x), u\rangle_x)u$ with high probability.

**Lemma 3.1.** *Assume $f$ is geodesically $L$-smooth. Let $u \sim \mathrm{Unif}(\mathcal{S}_{\mathrm{T}_x\mathcal{M}}(1))$, and $\nu \in (0, 1)$. Then with probability at least $1 - \gamma_x$, we have*

$$h_\nu(x) = \mathrm{sign}(\langle \mathrm{grad}f(x), u\rangle_x)u,$$

*where $h_\nu(x)$ is defined in (4) and*

$$\gamma_x = \sqrt{\frac{d}{2\pi}} \frac{L\nu}{\|\mathrm{grad}f(x)\|} \qquad (5)$$

Now in Lemma 3.2, setting $v = \mathrm{grad}f(x)$, we show that $\frac{\sqrt{d}}{\hat{C}}\mathrm{sign}(\langle \mathrm{grad}f(x), u\rangle_x)u$ is an unbiased estimator of the normalized gradient for some universal constant $\hat{C}$.

**Lemma 3.2.** *Fix $x \in \mathcal{M}$ and a nonzero vector $v \in \mathrm{T}_x\mathcal{M}$. Let $u \sim \mathrm{Unif}(\mathcal{S}_{\mathrm{T}_x\mathcal{M}}(1))$. The following holds for some universal constant $\hat{C} \in \left[\frac{1}{\sqrt{2\pi}}, 1\right]$:*

$$\mathbb{E}_u\big[\mathrm{sign}(\langle v, u\rangle_x)u\big] = \frac{\hat{C}}{\sqrt{d}} \frac{v}{\|v\|_x}. \qquad (6)$$

Using Lemma 3.1, and Lemma 3.2, Proposition 3.3 below shows that the estimator $h_\nu(x)$ is, on average, approximately aligned with the gradient direction.

**Proposition 3.3.** *Assume $f$ is geodesically $L$-smooth. For any $x \in \mathcal{M}$, $\nu \in (0, 1)$ and $v \in S_{\mathrm{T}_x\mathcal{M}}(1)$, it holds that*

$$\left| \mathbb{E}_u\big[\langle h_\nu(x), v\rangle\big] - \frac{\hat{C}}{\sqrt{d}} \left\langle \frac{\mathrm{grad}f(x)}{\|\mathrm{grad}f(x)\|}, v \right\rangle_x \right| \le 2\gamma_x,$$

*for some universal constant $\hat{C} \in [\frac{1}{\sqrt{2\pi}}, 1]$, where $\gamma_x$ is defined in (5).*

---

**Algorithm 1** Riemannian Dueling Normalized Gradient Descent (**RDNGD**)

---

1: Set $\hat{x}_0 = x_0$
2: **for** $k = 0, 1, \ldots, T - 1$ **do**
3:     Sample $u_k \sim \mathrm{Unif}(\mathcal{S}_{\mathrm{T}_{x_k}\mathcal{M}}(1))$
4:     $h_\nu(x_k) = \mathcal{Q}_f(\mathrm{Exp}_{x_k}(\nu u_k), \mathrm{Exp}_{x_k}(-\nu u_k))u_k$
5:     Update $x_{k+1} = \mathcal{P}_\mathcal{X}\left(\mathrm{Exp}_{x_k}(-\eta_k h_\nu(x_k))\right)$
6:     $\hat{x}_{k+1} = b_k\hat{x}_k + (1 - b_k)x_{k+1}$,
7:        where $b_k = (\mathcal{Q}(x_{k+1}, \hat{x}_k) + 1)/2$
8: **end for**
9: **Return** $\hat{x} := \hat{x}_T$

---

*Remark* 3.4. Setting $v = \mathrm{grad}f(x)/\|\mathrm{grad}f(x)\|$, and $\nu = \sqrt{2\pi}/d$, we see that $\sqrt{d}h_\nu(x)/\hat{C}$ is approximately aligned with gradient direction with error $O(L/(\hat{C}\|\mathrm{grad}f(x)\|))$. This bias is unavoidable as $\hat{C}$, and $\|\mathrm{grad}f(x)\|$ are unknown. This is a key distinction of our setting from standard stochastic Riemannian optimization algorithms where the latter relies on unbiased gradient estimators (Bonnabel, 2013; Tripuraneni et al., 2018). Lemma 3.2 improves the lower bound on $\hat{C}$ from $\frac{1}{20}$ in (Saha et al., 2021) to $\frac{1}{\sqrt{2\pi}} \approx 0.4$, leading to up to an eightfold reduction in gradient direction estimation error bound. Lemma 3.1 sharpens the bias bound by removing the logarithmic factor $\sqrt{\log\left(|\nabla f(\mathbf{x})|/(\sqrt{d}L\nu)\right)}$ present in $\gamma_x$ in (Saha et al., 2021) for the Euclidean case.

Building on the estimator $h_\nu(x)$, we now present our RDNGD method (Algorithm 1) for solving (1). Inputs of RDNGD are the initial point $x_0 \in \mathcal{X} \subseteq \mathcal{M}$, learning rate $\{\eta_k\}$, perturbation radius $\nu$, and maximum iteration number $T > 1$. $\hat{x}_k$ records the best iterate in the first $k$ iterations. The RDNGD method applies to two cases: (i) unconstrained problem where $f$ is geodesically $L$-smooth and $\mathcal{X} = \mathcal{M}$; (ii) constrained convex problem where $f$ is geodesically convex and geodesically $L$-smooth. The iteration complexities of RDNGD for obtaining an $\epsilon$-stationary solution in cases (i) and (ii) are presented in Theorem 3.6, and Theorem 3.7 respectively. For both cases, we consider two different choices of step sizes: the constant step size and the cosine annealing step size. The latter is defined as follows.

**Definition 3.5** (**Cosine Annealing step size**). Given the initial step size $\eta_0$, the minimum step size $\eta_{\min} \in [0, \eta_0]$, and the total number of iterations $T > 0$, the cosine annealing step size at the $k$-th iteration is defined as:

$$\eta_k = \eta_{\min} + \frac{1}{2}(\eta_0 - \eta_{\min})\left(1 + \cos\left(\frac{k\pi}{T}\right)\right) \qquad (7)$$

for $k = 0, 1, \ldots, T$.

**Theorem 3.6.** *Assume $f$ is geodesically $L$-smooth. We consider using RDNGD (Algorithm 1) to solve (1) with $\mathcal{X} = \mathcal{M}$[1]. For any given $\epsilon > 0$, RDNGD returns an $\epsilon$-*

---

[1]The projection operator in Algorithm 1 can be ignored here.

*stationary point with the following two sets of inputs.*

**(i) Constant step size.**

$$T = \left\lceil \frac{L^2 d(D+1)^2}{\hat{C}^2 \epsilon^2} \right\rceil, \; \eta_k \equiv \eta = \frac{1}{\sqrt{T}}, \; \nu = \frac{\hat{C}\sqrt{2\pi}}{4dL}\epsilon.$$

*In this case, RDNGD returns a point $x_R$ such that $\mathbb{E}[\|\mathrm{grad}f(x_R)\|] < \epsilon$ where $x_R$ is chosen uniformly at random from $\{x_0, \ldots, x_{T-1}\}$. Throughout this paper, $D := d^2(x_0, x^*)$ denotes the squared distance between the initial point and the optimal point.*

**(ii) Cosine annealing step size.**

$$T = \left\lceil \left(\frac{2\sqrt{d}}{\epsilon\hat{C}}\left(2LD + \tfrac{1}{2}L\right)\right)^2 \right\rceil, \quad \eta_0 = \frac{1}{\sqrt{T}},$$

$$\eta_k \text{ defined in (7)}, \quad \nu = \frac{\hat{C}\sqrt{\pi}}{2\sqrt{2}dL}\epsilon. \tag{8}$$

*In this case, RDNGD returns a point $x_R$ such that $\mathbb{E}[\|\mathrm{grad}f(x_R)\|] < \epsilon$ where $x_R$ is chosen at random from $\{x_0, \ldots, x_{T-1}\}$ with probability $\mathbb{P}(R = r) = \frac{\eta_r}{\sum_{k=1}^{T} \eta_k}$.*

**Theorem 3.7.** *Assume $f$ is geodesically $L$-smooth and geodesically convex, and Assumptions 2.5 and 2.6 hold. Consider using RDNGD (Algorithm 1) to solve (1) over a bounded set $\mathcal{X} \subset \mathcal{M}$ of diameter $\mathfrak{D}$. For any $\epsilon > 0$, RD-NGD returns an $\epsilon$-optimal solution satisfying $\mathbb{E}[f(\hat{x}_T)] - f(x^*) < \epsilon$ with the following two sets of inputs.*

**(i) Constant step size.**

$$T = \left\lceil 1 + \frac{16\pi dL\bar{\zeta}}{\epsilon}D \right\rceil, \quad \eta = \frac{\sqrt{\epsilon}}{4\sqrt{\pi}\bar{\zeta}\sqrt{dL}},$$

$$\nu = \frac{1}{4\sqrt{2}d}\frac{(\epsilon/L)^{3/2}}{(\sqrt{D} + \eta T)^2}. \tag{9}$$

*Here and in the rest of the paper, $\bar{\zeta} := \sqrt{|\kappa|}\mathfrak{D}/\tanh\left(\sqrt{|\kappa|}\mathfrak{D}\right)$, where $\kappa$ is the sectional curvature lower bound in Assumption 2.5.*

**(ii) Cosine annealing step size.**

$$T = \left\lceil 1 + \frac{32\pi dL\bar{\zeta}}{\epsilon}D \right\rceil, \quad \eta_0 = \frac{\sqrt{\epsilon}}{4\sqrt{\pi}\bar{\zeta}\sqrt{dL}},$$

$$\nu = \frac{1}{4\sqrt{2}d}\frac{(\epsilon/L)^{3/2}}{(\sqrt{D} + \frac{\eta_0}{2}T)^2}. \tag{10}$$

*Remark 3.8.* Here we remark that the Assumptions 2.5 and 2.6 naturally hold if $\mathcal{M}$ is a Hadamard manifold, and they are standard assumptions under such scenario. See (Bačák, 2014; Zhang & Sra, 2016; Li et al., 2024).

---

**Algorithm 2** Riemannian Recurrent Dueling Normalized Gradient Descent (**RRDNGD**)

1: **Input:** Initial point $x_0 \in \mathcal{X} \subset \mathcal{M}$, constant phase length $t = \left\lceil 1 + \frac{64\pi dL\bar{\zeta}}{\alpha} \right\rceil$, total number of phases $K$
2: **Initialize:** $x^{(0)} = x_0$, $D_0 = D$
3: **for** $k = 0, 1, \ldots, K-1$ **do**
4:     Run RDNGD with inputs $(x^{(k)}, \eta_k, \nu_k, t)$ to obtain $x^{(k+1)}$
5:     Update $D_{k+1} = D_k/2$
6: **end for**
7: **Return** $\hat{x} = x^{(K)}$

---

*Remark 3.9.* Our rates in Theorem 3.7 match the rates in (Saha et al., 2021) when the problem is reduced to the Euclidean case. However, (Saha et al., 2021) requires strictly smaller choices of step size to ensure convergence due to logarithmic factors in the analysis, whereas our analysis permits larger step size potentially leading to faster convergence in practice. Furthermore, when reduced to Euclidean case, our results in Theorems 3.6 and 3.7 also improve the dimension dependence in (Tao et al., 2026) by a factor of $\log d$. Additionally, our geometry-aware bounds are adaptive to the intrinsic manifold structure, in contrast to the ambient-dimension–dependent factors in prior analyses.

We now introduce a new algorithm that achieves linear convergence rate for solving (1) when $f$ is geodesically $\alpha$-strongly convex and geodesically $L$-smooth. The key observation is that under strong convexity, controlling $f(x) - f(x^*)$ directly controls the squared distance $d^2(x, x^*)$. We exploit this idea by designing an algorithm that runs in phases: in phase $k$, we run our RDNGD (Algorithm 1) to obtain an iterate with pre-given function value sub-optimality $\epsilon_k$, which leads to reductions in the estimation error. By reducing $\epsilon_k$ in each iteration, the convergence rate is improved to linear rate. The algorithm is called Riemannian Recurrent Dueling Normalized Gradient Descent (RRDNGD) and is presented in Algorithm 2, where we used a constant phase length $t = \lceil 1 + 64\pi dL\bar{\zeta}/\alpha \rceil$.

We now present the oracle complexity of RRDNGD.

**Theorem 3.10.** *Assume $f$ is geodesically $L$-smooth and geodesically $\alpha$-strongly convex. Assume Assumptions 2.5 and 2.6 hold, and the global optima $x^*$ lies in the interior of $\mathcal{X}$. Consider using RRDNGD (Algorithm 2) to solve (1) over a bounded set $\mathcal{X} \subset \mathcal{M}$ of diameter $\mathfrak{D}$. For any $\epsilon > 0$, we set $K = \left\lceil \log_2\left(\frac{l(L,\mathfrak{D})^2 D}{\epsilon^2}\right) \right\rceil$, where $l(L, \mathfrak{D})$ is constant dependent on $L$, and $\mathfrak{D}$. In the $k$-th iteration, set:*

$$\epsilon_k = \frac{\alpha D}{2^{2-k}}, \eta_k = \frac{1}{4\bar{\zeta}}\sqrt{\frac{\epsilon_k}{\pi dL}},$$

$$\nu_k = \frac{1}{4\sqrt{2}d}\frac{(\epsilon_k/L)^{3/2}}{(2\sqrt{\epsilon_k/\alpha} + \eta_k t)^2}.$$

**Algorithm 3** Riemannian Dueling Frank-Wolfe Method (**RDFW**)

---

1: **Input:** Initial point $x_0 \in \mathcal{X} \subseteq \mathcal{M}$, total number of iterations $T$
2: **for** $k = 0, 1, \ldots, T - 1$ **do**
3:    **for** $j = 1, \ldots, M_k$ **do**
4:       Sample $u_j^{(k)} \sim \text{Unif}(\mathcal{S}_{\text{T}_{x_k}}\mathcal{M}(1))$
5:       $o_j^{(k)} = \mathcal{Q}_f(\text{Exp}_{x_k}(\nu_k u_j^{(k)}), \text{Exp}_{x_k}(-\nu_k u_j^{(k)}))$
6:       $h_{\nu_k}^j(x_k) = o_j^{(k)} u_j^{(k)}$
7:    **end for**
8:    $\bar{h}_k \leftarrow \frac{1}{M_k} \sum_{j=1}^{M_k} h_{\nu_k}^j(x_k)$
9:    $z_k \leftarrow \arg\min_{z \in \mathcal{X}} \langle \bar{h}_k, \text{Log}_{x_k}(z) \rangle$
10:   $s_k \leftarrow \frac{2}{k+3}$
11:   $x_{k+1} = \text{Exp}_{x_k}(s_k \text{Log}_{x_k}(z_k))$
12: **end for**
13: **Return** $\hat{x} = x_T$

---

*Then RRDNGD returns an $\epsilon$-optimal solution $\hat{x}$ satisfying $\mathbb{E}[f(\hat{x})] - f(x^*) \leq \epsilon$ with an oracle complexity of $\mathcal{O}\left(\frac{dL\bar{\zeta}}{\alpha} \log\left(\frac{l(L,\mathfrak{D})\sqrt{D}}{\epsilon}\right)\right)^2$.*

## 4. Riemannian Dueling Frank-Wolfe Method

Note that RDNGD method requires a projection in each iteration, which can be expensive for some applications. To overcome this issue, in this section, we design a projection-free Riemannian dueling Frank-Wolfe (RDFW) method using comparison oracles. In the Euclidean space, the Frank-Wolfe method calls a linear minimization oracle in each iteration. On manifolds, at iteration $k$, Frank-Wolfe method requires minimizing the following analogous subproblem:

$$\arg\min_{z \in \mathcal{X}} \langle \text{grad} f(x_k), \text{Log}_{x_k}(z) \rangle. \tag{11}$$

In some applications, unlike projection, (11) admits a closed–form solution. For example, consider the manifold of symmetric positive definite (SPD) matrices $\mathcal{M} = \mathcal{S}_{++}^n := \{X \in \mathbb{R}^{n \times n} \mid X^T = X, X \succ 0\}$ with a geodesically convex "interval" constraint $\mathcal{X} := \{X \in \mathcal{M} : L \preceq X \preceq U\}$ for SPD matrices $L$ and $U$. When $L$ and $U$ do not commute, the Riemannian projection requires an expensive iterative solver. In contrast, (11) admits a closed–form solution that is cheap to compute (Weber & Sra, 2023).

We now present our RDFW algorithm with the detailed description in Algorithm 3. In the absence of gradient information in dueling feedback setting, RDFW solves the subproblem (11) with $\text{grad} f(x)$ replaced by our gradient direction estimator. However, the solution to (11) is highly

---

[2]Our complexity improves the dependency on the initial squared distance from linear in (Saha et al., 2021), which is $\mathcal{O}\left(\frac{dL}{\alpha}(\log \frac{\alpha}{\epsilon} + \|\mathbf{x}_1 - \mathbf{x}^*\|^2)\right)$, to logarithmic.

sensitive to the noise in gradient estimation (see Section H). To reduce the noise variance, in the $k$-th iteration, we use the batch gradient direction estimator given by:

$$\bar{h}_k = \frac{1}{M_k} \sum_{j=1}^{M_k} o_j^{(k)} u_j^{(k)}, \tag{12}$$

where we use $M_k$ i.i.d. directions $u_j^{(k)} \sim \text{Unif}(\mathcal{S}_{\text{T}_{x_k}}\mathcal{M}(1))$. The variance of $\bar{h}_k$ reduces at a $\frac{1}{M_k}$ rate. We now solve the following Riemannian Frank–Wolfe subproblem:

$$z_k \in \arg\min_{z \in \mathcal{X}} \langle \bar{h}_k, \text{Log}_{x_k}(z) \rangle.$$

The update $x_{k+1}$ is obtained by moving along the geodesic from $x_k$ to $z_k$, and this step preserves feasibility. Theorem 4.1 gives the oracle complexity of RDFW.

**Theorem 4.1.** *Assume $f$ is geodesically $L$-smooth and geodesically convex. For any given $\epsilon > 0$, suppose RDFW (Algorithm 3) is run with*

$$\nu_k = \sqrt{\frac{\pi}{8}} \frac{\hat{C}}{dL\mathfrak{D}} \frac{2}{k+3}, \quad M_k = \frac{8dL\mathfrak{D}^2}{\hat{C}^2}(k+3),$$
$$T = \left\lceil \frac{2(f(x_0) - f(x^*)) + 4L\mathfrak{D}^2 + 12}{\epsilon} \right\rceil,$$

*then $\mathbb{E}\left[f(x_T) - f(x^*)\right] < \epsilon$, and the oracle complexity is $2\sum_{k=0}^{T-1} M_k = \mathcal{O}(d/\epsilon^2)$.*

## 5. Numerical Experiments

In this section, we conduct numerical experiments to test our algorithms on both synthetic data and real applications.

### 5.1. Synthetic Problems

We consider three problems with synthetic data: Rayleigh quotient maximization, the Karcher mean problem, and the Karcher mean problem with an additional constraint.

#### 5.1.1. RAYLEIGH QUOTIENT MAXIMIZATION

Given a symmetric matrix $A \in \mathbb{R}^{d \times d}$, the Rayleigh quotient maximization problem is formulated as the minimization of the negative quadratic form over the unit sphere:

$$\min_{x \in \mathcal{S}_d(1)} f(x) := -\frac{1}{2} x^\top A x. \tag{13}$$

The objective $f$ is geodesically $L$-smooth with $L = \lambda_{\max}(A) - \lambda_{\min}(A)$ (Kim & Yang, 2022) and $f^* = -\frac{1}{2}\lambda_{\max}(A)$. Following the setup in (Kim & Yang, 2022), we generate a random matrix $B \in \mathbb{R}^{d \times d}$ with entries sampled i.i.d. from $\mathcal{N}(0, 1/d)$ and set $A = \frac{1}{2}(B + B^\top)$. We compare our RDNGD (Algorithm 1) with the zeroth-order Riemannian gradient descent method (ZO-RGD) (Li

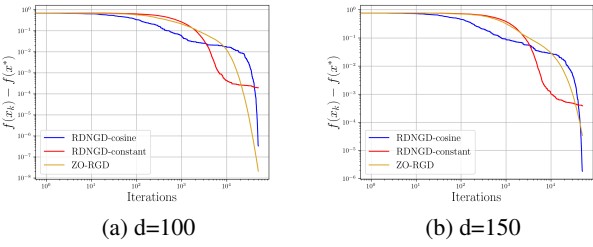

(a) d=100      (b) d=150

*Figure 1.* Numerical results for Rayleigh quotient maximization.

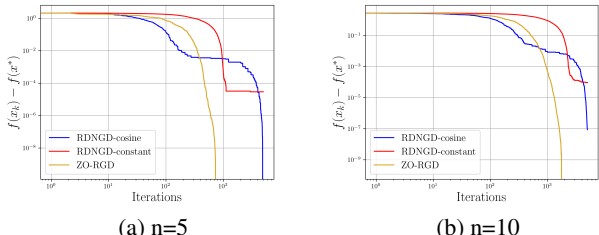

(a) n=5      (b) n=10

*Figure 2.* Numerical result for Karcher mean problem.

et al., 2023). While ZO-RGD requires function values, RDNGD relies only on comparison oracles yet achieves comparable performance. The algorithms are initialized at $x_0 \sim \mathrm{Unif}(\mathcal{S}_d(1))$. RDNGD-constant uses a constant step size $\eta = 10^{-2}$, and RDNGD-cosine employs cosine annealed step size with $\eta_0 = 10^{-1}$ and $\eta_{\min} = 10^{-8}$. Both RDNGD variants use $\nu = 10^{-8}$. For ZO-RGD (Li et al., 2023), we set the smoothing parameter as $10^{-6}$ and the step size as $\frac{1}{2dL}$. These values are tuned via a grid search. All algorithms are run for $T = 50,000$ iterations.

Figure 1 reports $f(x_k) - f(x^*)$ for different methods for (13) for dimensions $d = 100$ and $d = 150$. We see that RDNGD-cosine achieves similar performance as ZO-RGD although the former uses only the comparison oracle. Although RDNGD-constant performed worse than RDNGD-cosine and ZO-RGD but still achieves acceptable accuracy.

### 5.1.2. KARCHER MEAN PROBLEM

In this problem, the goal is to find the geometric mean or Karcher mean of $m$ SPD matrices $\{A_i\}_{i=1}^m \subset \mathcal{S}_{++}^n$, and can be cast as follows:

$$\min_{X \in \mathcal{S}_{++}^n} f(X) := \frac{1}{2m} \sum_{i=1}^m d^2(X, A_i), \qquad (14)$$

where $d(\cdot, \cdot)$ is the Riemannian distance induced by the affine-invariant metric on $\mathcal{S}_{++}^n$. $f$ in (14) is geodesically $L$-smooth and geodesically convex on $\mathcal{S}_{++}^n$ (Zhang & Sra, 2016). Here we again compare our RDNGD algorithm with the ZO-RGD algorithm. The matrices $A_i$ were randomly generated using the Matrix Mean Toolbox (Bini & Iannazzo, 2013) with a condition number of $10^6$ and sample size $m = 50$. We consider dimensions $n \in \{5, 10\}$. The algorithms are initialized at $X_0$, which is set to the arithmetic mean of all $A_i$'s. To evaluate the performance, the optimal solution $X^*$ of (14) is computed using the Richardson-like linear gradient descent algorithm provided in the Matrix Mean Toolbox. RDNGD-constant uses a constant step size $\eta = 10^{-2}$, and RDNGD-cosine employs $\eta_0 = 10^{-1}$ and $\eta_{\min} = 10^{-6}$. Both RDNGD variants use $\nu = 10^{-8}$. For ZO-RGD (Li et al., 2023), we set the smoothing parameter as $10^{-6}$ and the step size as $\frac{1}{20nL}$ and these values are again tuned

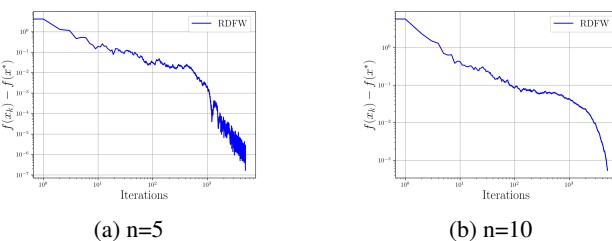

(a) n=5      (b) n=10

*Figure 3.* Numerical result for constrained Karcher mean problem.

via a grid search to produce the best results for this problem. All algorithms are run for $T = 5,000$ iterations.

Figure 2 reports the convergence of different methods on (14) for dimensions $n \in \{5, 10\}$. We see that RDNGD-cosine can achieve comparable accuracy with ZO-RGD. RDNGD-constant performed slightly worse but the accuracy is still on an acceptable level. This is again remarkable because RDNGD can only access a comparison oracle.

### 5.1.3. KARCHER MEAN WITH CONSTRAINTS

In this subsection, we apply our RDFW method (Algorithm 3) to solve the Karcher mean problem with the constraint $H \preceq X \preceq A$ where $H$ is the harmonic mean, and $A$ is the arithmetic mean (Weber & Sra, 2023). It is a natural constraint as the Karcher mean $G$ satisfies $H \preceq G \preceq A$ (Bhatia, 2007). Formally, we obtain the following formulation:

$$\min_{X \in \mathcal{S}_{++}^n, \ H \preceq X \preceq A} f(X) := \frac{1}{2m} \sum_{i=1}^m d^2(X, A_i).$$

We set $m = 50$ and consider dimensions $n \in \{5, 10\}$. The algorithm is initialized at $X_0 = A$. We set the perturbation parameter to $\nu_k = 10^{-8}$, the batch size to $M_k = \lceil (k + 1)/500 \rceil$, and the total number of iterations to $T = 5,000$.

Figure 3 shows that RDFW converges to high-accuracy solutions for the constrained Karcher mean problem. To our knowledge, RDFW is the first projection-free algorithm to solve this problem using only comparison oracles.

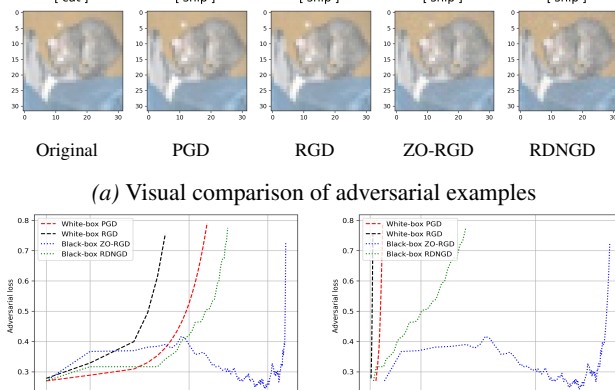

*(a)* Visual comparison of adversarial examples

*(b)* Adversarial loss vs. Iteration  *(c)* Adversarial loss vs. time

*Figure 4.* Attack results on CIFAR-10. (a) Generated adversarial images. (b) and (c) show the convergence curves. Our estimator yields accurate gradient estimation with only 10 samples (vs. 500 for ZO-RGD), demonstrating superior query efficiency.

### 5.2. Real Applications

#### 5.2.1. ATTACK ON DEEP NEURAL NETWORK

We evaluate our dueling Riemannian attack on the black-box benchmark (Li et al., 2023), attacking a VGG network on CIFAR-10 under an $\ell_2$-norm constraint. The perturbation is confined to a sphere (dimension $\approx 1000$) centered at the original image $x \in \mathbb{R}^d$. We adopt the exact experimental settings from (Li et al., 2023).[3] For our RDNGD algorithm, we set $\nu = 10^{-6}$, step size $\eta = 10^{-6}$, and $T = 1,000$. We use the batch gradient direction estimator $\bar{h}_k$ (12) with batch size of 10 to mitigate the high variance of the gradient direction estimator in such a high-dimensional space. In contrast, ZO-RGD request 500 samples.

Figure 4 illustrates representative adversarial examples (additional results in Section I) generated by different attacks. The white-box methods PGD and RGD are included as references but are infeasible in our comparison oracle setting. For all these test images, the attacks successfully induce misclassification with visually indistinguishable perturbations to the original image. Among the practical black-box methods, ZO-RGD and our RDNGD attack, the loss curves show that RDNGD yields a more stable optimization trajectory and reaches higher adversarial loss in fewer iterations and less CPU time, indicating a clear efficiency advantage. Moreover, as noted earlier, RDNGD operates with strictly weaker oracle information than ZO-RGD which makes its performance gains even more remarkable.

---

[3]All attacks are implemented by extending the open-source codebase available at https://github.com/JasonJiaxiangLi/Zeroth-order-Riemannian

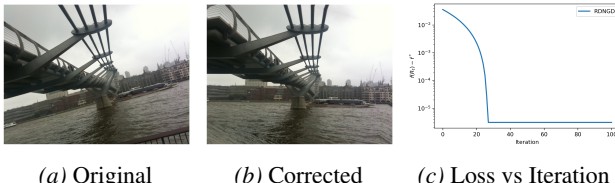

*(a)* Original  *(b)* Corrected  *(c)* Loss vs Iteration

*Figure 5.* Horizon leveling results. The loss stabilizes at $10^{-5}$ due to a large constant step size, chosen to balance convergence speed and the optimality gap.

#### 5.2.2. HORIZON LEVELING

In this experiment, we evaluate our methods in the horizon leveling problem presented in Section 1.2. For each image in the HLW dataset (Workman et al., 2016), we compute $R_{\text{tilt}} \in \mathrm{SO}(2)$ between the human-annotated horizon and the horizontal axis. Our goal is to find the correction rotation $R^\star \in \mathrm{SO}(2)$ that minimizes the misalignment cost $f(R)$:

$$\min_{R \in \mathrm{SO}(2)} f(R) := \|RR_{\text{tilt}} - I\|_F^2.$$

While dueling feedback could be obtained from human comparisons, we use $f(R)$ as a scalable and reproducible surrogate for human preference. Given two rotations $R_1$ and $R_2$, the oracle returns the one with smaller misalignment, providing pairwise preferences on $\mathrm{SO}(2)$ without revealing function values. We set the maximum iteration number $T = 100$, $\nu = 10^{-6}$, and constant step size $\eta = 10^{-2}$.

Figure 5 shows an example of our dueling horizon-leveling procedure on real image (additional results in Section I). Across all cases, suboptimality reaches $\sim 10^{-5}$ within 30 iterations, showing high accuracy with few iterations despite no access to function values or gradients. In each example, the corrected image looks visually level indicating that the algorithm successfully recovers a good rotation.

## 6. Conclusion

In this paper, we establish the first theoretical framework for Riemannian dueling optimization that uses only comparison oracles. We proposed three algorithms for tackling Riemannian dueling optimization in different scenarios: (i) geodesically $L$-smooth objective; (ii) geodesically $L$-smooth and geodesically (strongly) convex objective where projection onto the constraint set $\mathcal{X}$ is cheap; (iii) geodesically $L$-smooth and geodesically convex objective when projection is prohibited. We established iteration and oracle complexities for each scenario. Moreover, our refined analysis overcomes intrinsic geometric barriers and improves upon existing Euclidean results by offering tighter constants and more flexible hyperparameter selection. Numerical experimental results on both synthetic problems and real applications demonstrated the capability of the proposed algorithms. Our work opens several promising directions on

Riemannian optimization with dueling feedback, e.g., accelerated algorithms, Hessian-aware methods for identifying local minima, and exploring whether the noise in direction estimator can facilitate escape from saddle points.

## Acknowledgements

Shiqian Ma is partly supported by ONR grant N00014-24-1-2705, NSF grants CCF-2311275 and ECCS-2326591.

## Impact Statement

This paper presents work whose goal is to advance the field of Machine Learning. There are many potential societal consequences of our work, none which we feel must be specifically highlighted here.

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

# A. Preliminaries on Riemannian Geometry

Let $\mathcal{M}$ be a Riemannian manifold. We introduce some relevant geometric concepts and definitions below.

**Definition A.1** (**Tangent Vector and Tangent Space**). Let $\mathcal{M}$ be a Riemannian manifold and let $x$ be a point on the manifold $\mathcal{M}$. Define $\mathfrak{F}_x(\mathcal{M})$ to be the set of smooth functions on $\mathcal{M}$ defined on the neighborhood of $x \in \mathcal{M}$. A tangent vector $v_x : \mathfrak{F}_x(\mathcal{M}) \to \mathbb{R}$ at $x$ is a linear operator such that

$$v_x(f) = \left. \frac{d(f(\gamma(t)))}{dt} \right|_{t=0}, f \in \mathfrak{F}(\mathcal{M}),$$

where $\gamma : (-\delta, \delta) \to \mathcal{M}$ is a smooth curve on $\mathcal{M}$ with $\gamma(0) = x$. The tangent space $\mathrm{T}_x\mathcal{M}$ is the linear space of all tangent vector at $x$.

**Definition A.2** (**Riemannian Gradient**). Suppose $f$ is a smooth function on $\mathcal{M}$. The Riemannian gradient $\mathrm{grad}f(x)$ is the unique tangent vector in $\mathrm{T}_x\mathcal{M}$ satisfying

$$\langle v_x, \mathrm{grad}f(x) \rangle_x = v_x(f) \quad \forall v_x \in \mathrm{T}_x\mathcal{M}.$$

Geodesics extend the conceppt of straight line to manifold, and represents the locally distance-minimizing paths.

**Definition A.3** (**Geodesic**). Let $(\mathcal{M}, g)$ be a Riemannian manifold. A smooth curve $\gamma : I \to \mathcal{M}$, where $I \subset \mathbb{R}$ is an interval, is called a geodesic if it satisfies that

$$\nabla_{\dot{\gamma}(t)}\dot{\gamma}(t) = 0 \quad \text{for all } t \in I,$$

where $\nabla$ is the Levi-Civita connection associated with the metric $g$.

The exponential mapping at a point $x \in M$ maps a tangent vector $v \in \mathrm{T}_x M$ to the point reached by the unique geodesic starting at $x$ with initial velocity $v$, evaluated at time $t = 1$.

**Definition A.4** (**Exponential Mapping**). Let $\mathcal{M}$ be a Riemannian manifold and $x \in M$. The exponential mapping at $x$, denoted $\mathrm{Exp}_x : \mathrm{T}_x\mathcal{M} \to \mathcal{M}$, is defined by

$$\mathrm{Exp}_x(v) = \gamma_v(1),$$

where $\gamma_v : [0, 1] \to \mathcal{M}$ is the unique geodesic such that $\gamma_v(0) = x$ and $\dot{\gamma}_v(0) = v \in \mathrm{T}_x\mathcal{M}$. That is, $\mathrm{Exp}_x(v)$ is the endpoint of the geodesic starting at $x$ with initial velocity $v$, evaluated at time $t = 1$.

**Definition A.5** (**Riemannian distance**). Let $(\mathcal{M}, g)$ be a Riemannian manifold with metric tensor $g$. For any two points $x, y \in \mathcal{M}$, the Riemannian distance between $x$ and $y$ is defined as

$$d(x, y) := \inf_{\gamma \in \Gamma(x,y)} \int_0^1 \sqrt{g_{\gamma(t)}\big(\dot{\gamma}(t), \dot{\gamma}(t)\big)} dt,$$

where $\Gamma(x, y)$ denotes the set of all smooth curves $\gamma : [0, 1] \to \mathcal{M}$ such that $\gamma(0) = x$ and $\gamma(1) = y$.

*Remark* A.6. The exponential mapping $\mathrm{Exp}_x$ is a local diffeomorphism. That is, within a open neighborhood (restricted by the manifold), $\mathrm{Exp}_x$ is invertible and well defined. We use $\mathrm{Log}_x$ to denote the invert map $\mathrm{Exp}_x^{-1}$.

Tangent vectors at two different points lie in different spaces, and thus cannot be directly compared. To solve this issue, parallel transport is defined to move a tangent vector along the geodesics in the meanwhile preserving the length and direction.

**Definition A.7** (**Parallel Transport**). Let $\gamma : [0, 1] \to \mathcal{M}$ be a smooth curve on a Riemannian manifold $\mathcal{M}$, and let $V(t) \in T_{\gamma(t)}\mathcal{M}$ be a vector field along $\gamma$. The vector field $V$ is said to be parallel along $\gamma$ if it satisfies

$$\nabla_{\dot{\gamma}(t)}V(t) = 0 \quad \text{for all } t \in [0, 1].$$

Given an initial vector $v_0 \in T_{\gamma(0)}\mathcal{M}$, there exists a unique parallel vector field $V(t)$ such that $V(0) = v_0$. The vector $V(1) \in T_{\gamma(1)}\mathcal{M}$ is called the parallel transport of $v_0$ along $\gamma$.

# B. Some Useful Lemmas

In the proof provided in the appendix, we adopt the following notation: for any positive integer $N$, let $[N]$ denote the set $\{0, 1, \ldots, N\}$. Additionally, we define $\mathcal{U}_k = \{u_0, u_1, \ldots, u_k\}$ as the history containing all information available up to step $k$.

**Lemma B.1.** *Let $\{\eta_k\}_{k=0}^{T-1}$ be the cosine annealing step size defined in Definition 3.5[4]. It holds that:*

- $\displaystyle\sum_{k=0}^{T-1} \cos\left(\frac{\pi k}{T}\right) = 1.$

- $\displaystyle\sum_{k=0}^{T-1} \eta_k = \frac{\eta_0}{2}\left(T + \sum_{k=0}^{T-1} \cos\left(\frac{\pi k}{T}\right)\right) = \frac{\eta_0}{2}(T+1).$

- $\displaystyle\sum_{k=0}^{T-1} \eta_k^2 = \frac{\eta_0^2}{8}(3T+4).$

*Proof.* We first prove part (i). According to (Li et al., 2021), it holds that $\sum_{k=1}^{T} \cos\left(\frac{k\pi}{T}\right) = -1$. Adjusting the summation range to $k = 0, \ldots, T-1$, we obtain

$$\sum_{k=0}^{T-1} \cos\left(\frac{k\pi}{T}\right) = \sum_{k=1}^{T} \cos\left(\frac{k\pi}{T}\right) - \cos(\pi) + \cos(0) = -1 - (-1) + 1 = 1.$$

Part (ii) follows immediately by substituting the result of part (i) into the definition of $\eta_k$:

$$\sum_{k=0}^{T-1} \eta_k = \frac{\eta_0}{2}\left(T + \sum_{k=0}^{T-1} \cos\left(\frac{\pi k}{T}\right)\right) = \frac{\eta_0}{2}(T+1).$$

For part (iii), expanding the square of $\eta_k$ yields

$$\sum_{k=0}^{T-1} \eta_k^2 = \frac{\eta_0^2}{4} \sum_{k=0}^{T-1}\left(1 + 2\cos\left(\frac{\pi k}{T}\right) + \cos^2\left(\frac{\pi k}{T}\right)\right).$$

Using the identity $\cos^2(x) = \frac{1+\cos(2x)}{2}$ and the fact that $\sum_{k=0}^{T-1} \cos\left(\frac{2\pi k}{T}\right) = 0$, we have

$$\sum_{k=0}^{T-1} \cos^2\left(\frac{\pi k}{T}\right) = \sum_{k=0}^{T-1} \frac{1}{2} + \frac{1}{2}\sum_{k=0}^{T-1} \cos\left(\frac{2\pi k}{T}\right) = \frac{T}{2}.$$

Substituting this result and the conclusion from part (i) back into the expansion gives

$$\sum_{k=0}^{T-1} \eta_k^2 = \frac{\eta_0^2}{4}\left(T + 2(1) + \frac{T}{2}\right) = \frac{\eta_0^2}{8}(3T+4).$$

This completes the proof. $\square$

**Lemma B.2.** *[(Zhang & Sra, 2016) Corollary 8] For any Riemannian manifold $\mathcal{M}$ where the sectional curvature is lower bounded by $\kappa$, the following holds for any $x, x_k \in \mathcal{X}$ and $x_{k+1} = \mathcal{P}_{\mathcal{X}}(\mathrm{Exp}_{x_k}(-\eta_k g_k))$:*

$$\langle -g_k, \mathrm{Log}_{x_k}(x)\rangle \le \frac{1}{2\eta_k}(d^2(x_k, x) - d^2(x_{k+1}, x)) + \frac{\zeta(\kappa, d(x_k, x))\eta_k}{2}\|g_k\|^2,$$

*where $\zeta(\kappa, d(x_k, x)) = \frac{\sqrt{|\kappa|}d(x_k,x)}{\tanh\left(\sqrt{|\kappa|}d(x_k,x)\right)}$.*

---

[4]Here, similar to (Li et al., 2021), we assume $\eta_{\min} = 0$ for simplicity.

**Lemma B.3.** *Suppose $f : \mathcal{M} \to \mathbb{R}$ is geodesically $L$-smooth. Consider the updates rule in Algorithm 1. Then we have for all $k \in [T]$,*

$$d(x_k, x^*) \leq \sqrt{D} + \sum_{t=0}^{k-1} \eta_t \quad and \quad \|\mathrm{grad}f(x_k)\| \leq L \left( \sqrt{D} + \sum_{t=0}^{k-1} \eta_t \right).$$

*Proof.* Using non-expansiveness of the projection operator, we have that

$$
\begin{aligned}
d(x_k, x^*) &\leq \sum_{t=0}^{k-1} d(x_{t+1}, x_t) + d(x_0, x^*) \\
&= \sum_{t=0}^{k-1} d(\mathcal{P}_{\mathcal{X}}(\mathrm{Exp}_{x_t}(-\eta_t h_t)), \mathcal{P}_{\mathcal{X}}(x_t)) + d(x_0, x^*) \\
&\leq \sum_{t=0}^{k-1} d(\mathrm{Exp}_{x_t}(-\eta_t h_t), x_t) + d(x_0, x^*) \\
&\leq \sum_{t=0}^{k-1} \eta_t + \sqrt{D}.
\end{aligned}
$$

From that $f$ is geodesically $L$-smooth, we have

$$\|\mathrm{grad}f(x_k)\| = \|\mathrm{grad}f(x_k) - \Gamma_{x^*}^{x_k} \mathrm{grad}f(x^*)\| \leq Ld(x_k, x^*) \leq L \left( \sqrt{D} + \sum_{t=0}^{k-1} \eta_t \right).$$

This completes the proof. $\qquad\square$

**Lemma B.4.** *Suppose $f : \mathcal{M} \to \mathbb{R}$ is a geodesically convex function. Then $f(x_k) - f(x^*) > \epsilon$ implies $\|\mathrm{grad}f(x_k)\| \geq \epsilon/d(x_k, x^*)$.*

*Proof.* By geodesically convexity, we get

$$f(x^*) \geq f(x_k) + \langle \mathrm{grad}f(x_k), \mathrm{Log}_{x_k}(x^*) \rangle.$$

Using the Cauchy-Schwarz inequality and that $d(x_k, x^*) = \|\mathrm{Log}_{x_k}(x^*)\|$, we obtain

$$\epsilon < f(x_k) - f(x^*) \leq -\langle \mathrm{grad}f(x_k), \mathrm{Log}_{x_k}(x^*) \rangle \leq \|\mathrm{grad}f(x_k)\| \cdot d(x_k, x^*).$$

Dividing both sides by $d(x_k, x^*)$ completes the proof. $\qquad\square$

**Lemma B.5.** *Let $f : \mathcal{X} \subseteq \mathcal{M} :\to \mathbb{R}$ be a geodesically convex and geodesically $L$-smooth function, and $x^*$ be its minimizer over $\mathcal{X}$. For any $x_k \in \mathcal{X}$ such that $f(x_k) - f(x^*) \geq \epsilon$, we have*

$$\left\langle \frac{\mathrm{grad}f(x_k)}{\|\mathrm{grad}f(x_k)\|}, \frac{\mathrm{Log}_{x_k}(x^*)}{\|\mathrm{Log}_{x_k}(x^*)\|} \right\rangle \leq -\frac{1}{\|\mathrm{Log}_{x_k}(x^*)\|} \sqrt{\frac{\epsilon}{2L}}.$$

*Proof.* From geodesically convexity, we have $f(x^*) \geq f(x_k) + \langle \mathrm{grad}f(x_k), \mathrm{Log}_{x_k}(x^*) \rangle_{x_k}$. Denote $g_k := \mathrm{grad}f(x_k)$ and $w_k := \frac{g_k}{\|g_k\|}$. This inequality can be rearranged as

$$\left\langle w_k, \mathrm{Log}_{x_k}(x^*) \right\rangle_{x_k} \leq -\frac{f(x_k) - f(x^*)}{\|g_k\|}. \tag{15}$$

Next, from geodesically $L$-smoothness, for any tangent vector $u \in \mathrm{T}_{x_k}\mathcal{M}$, we have

$$f(\mathrm{Exp}_{x_k}(u)) \leq f(x_k) + \langle g_k, u \rangle_{x_k} + \frac{L}{2}\|u\|^2.$$

Setting $u = -g_k/L$ and noting that $f(x^*) \le f(\text{Exp}_{x_k}(u))$, we get $f(x^*) \le f(x_k) - \frac{1}{2L}\|g_k\|^2$. and thus $\|g_k\| \le \sqrt{2L\left(f(x_k) - f(x^*)\right)}$, which, combining with (15) and $f(x_k) - f(x^*) \ge \epsilon$, yields

$$\left\langle w_k, \text{Log}_{x_k}(x^*)\right\rangle_{x_k} \le -\frac{f(x_k) - f(x^*)}{\sqrt{2L(f(x_k) - f(x^*))}} = -\sqrt{\frac{f(x_k) - f(x^*)}{2L}} \le -\sqrt{\frac{\epsilon}{2L}}.$$

Finally, dividing both sides by $\|\text{Log}_{x_k}(x^*)\|$ yields the desired result. $\qquad\square$

**Lemma B.6.** *Let $f : \mathcal{X} \subseteq \mathcal{M} \to \mathbb{R}$ be a geodesically $L$-smooth function, and let $x^*$ be its minimizer over $\mathcal{X}$. For any $x \in \mathcal{X}$, we have*
$$\|\text{grad}f(x)\|^2 \le 2L(f(x) - f(x^*)).$$

*Proof.* By the definition of geodesic $L$-smoothness, for any tangent vector $u \in \text{T}_x\mathcal{M}$, we have

$$f(\text{Exp}_x(u)) \le f(x) + \langle \text{grad}f(x), u\rangle_x + \frac{L}{2}\|u\|^2.$$

Consider the update direction $u = -\frac{1}{L}\text{grad}f(x)$. Plugging this into the inequality yields

$$f(\text{Exp}_x(u)) \le f(x) - \frac{1}{L}\|\text{grad}f(x)\|^2 + \frac{L}{2}\left\|-\frac{1}{L}\text{grad}f(x)\right\|^2$$
$$= f(x) - \frac{1}{2L}\|\text{grad}f(x)\|^2.$$

Since $x^*$ is the global minimizer, we have $f(x^*) \le f(\text{Exp}_x(u))$. Combining this with the upper bound derived above implies

$$f(x^*) \le f(x) - \frac{1}{2L}\|\text{grad}f(x)\|^2.$$

Rearranging the terms completes the proof. $\qquad\square$

## C. Proof of Lemmas 3.1, 3.2, and Proposition 3.3

### C.1. Proof of Lemma 3.1

*Proof.* From Definition 2.1, we have

$$\begin{array}{ccccc}
\langle \nu u, \text{grad}f(x)\rangle - \frac{L}{2}\nu^2 & \le & f(\text{Exp}_x(\nu u)) - f(x) & \le & \langle \nu u, \text{grad}f(x)\rangle + \frac{L}{2}\nu^2, \\
-\langle \nu u, \text{grad}f(x)\rangle - \frac{L}{2}\nu^2 & \le & f(\text{Exp}_x(-\nu u))) - f(x) & \le & -\langle \nu u, \text{grad}f(x)\rangle + \frac{L}{2}\nu^2,
\end{array}$$

which further implies

$$2\langle \nu u, \text{grad}f(x)\rangle - L\nu^2 \le f(\text{Exp}_x(\nu u)) - f(\text{Exp}_x(-\nu u))) \le 2\langle \nu u, \text{grad}f(x)\rangle + L\nu^2.$$

Note that this indicates if $|\langle u, \text{grad}f(x)\rangle| > L\nu/2$, then

$$\text{sign}(f(\text{Exp}_x(\nu u)) - f(\text{Exp}_x(-\nu u))) = \text{sign}(\langle u, \text{grad}f(x)\rangle). \tag{16}$$

We now establish an upper bound on $\mathbb{P}(|\langle u, \text{grad}f(x)\rangle| \le L\nu/2)$. Denote $a = \frac{\text{grad}f(x)}{\|\text{grad}f(x)\|}$, $b = \frac{L\nu}{2\|\text{grad}f(x)\|}$, and $Z = \langle u, a\rangle$. Therefore, $\mathbb{P}(|\langle u, \text{grad}f(x)\rangle| \le L\nu/2) = \mathbb{P}(|Z| \le b)$. By the rotational symmetry of the sphere, we can assume $a = e_1$, which implies $Z = u_1$, the first coordinate of $u$.

Similar to the proof of Lemma C.2, the marginal density of $Z \in [-1, 1]$ is,

$$f_Z(z) = A_d(1 - z^2)^{(d-3)/2} \quad \text{for } z \in [-1, 1], \text{ where } A_d = \frac{\Gamma\left(\frac{d}{2}\right)}{\sqrt{\pi}\Gamma\left(\frac{d-1}{2}\right)}.$$

Since $(1 - z^2)^{(d-3)/2} \leq 1$ for all $z \in [0, 1]$, we have $\int_0^b f_Z(z)dz \leq A_d b$, which further implies,

$$\mathbb{P}(|Z| \leq b) = \mathbb{P}(|\langle u, a \rangle| \leq b) = \int_{-b}^b f_Z(z)dz \leq 2A_d b \leq 2\sqrt{\frac{d}{2\pi}} b = \sqrt{\frac{d}{2\pi}} \frac{L\nu}{\|\mathrm{grad} f(x)\|},$$

where the second inequality is from the Gautschi's inequality. Therefore, from (16) we have that

$$\mathbb{P}(\mathrm{sign}(f(\mathrm{Exp}_x(\nu u)) - f(\mathrm{Exp}_x(-\nu u))) = \mathrm{sign}(\langle u, \mathrm{grad} f(x) \rangle))$$

$$\geq \mathbb{P}(|\langle u, \mathrm{grad} f(x) \rangle| > L\nu/2) \geq 1 - \sqrt{\frac{d}{2\pi}} \frac{L\nu}{\|\mathrm{grad} f(x)\|}.$$

This completes the proof. $\qquad\qquad\qquad\qquad\qquad\qquad\qquad\qquad\qquad\qquad\qquad\qquad\qquad\quad$ $\square$

### C.2. Proof of Lemma 3.2

*Proof.* Note that,

$$\mathbb{E}[\mathrm{sign}(\langle v, u \rangle_x)u] = \mathbb{E}\left[\mathrm{sign}\left(\left\langle \frac{v}{\|v\|_x}, u \right\rangle_x\right)u\right].$$

So, we can assume $\|v\|_x = 1$ without loss of generality. Let $P : \mathrm{T}_x\mathcal{M} \to \mathrm{T}_x\mathcal{M}$ denote the reflection operator along $v$ on $\mathrm{T}_x\mathcal{M}$, i.e., $P(\xi) = 2\langle v, \xi \rangle v - \xi$. For an arbitrary unit tangent vector $u \in \mathrm{T}_x\mathcal{M}$, denote $u' := P(u)$. We have that

$$\mathrm{sign}(\langle v, u' \rangle) = \mathrm{sign}\left(2\|v\|^2\langle v, u \rangle - \langle v, u \rangle\right) = \mathrm{sign}(\langle v, u \rangle).$$

Since $u' \sim \mathrm{Unif}(\mathcal{S}_{\mathrm{T}_x\mathcal{M}}(1))$, we then have

$$\begin{aligned}
\mathbb{E}[\mathrm{sign}(\langle v, u \rangle)u] &= \frac{1}{2}\mathbb{E}[\mathrm{sign}(\langle v, u \rangle)u] + \frac{1}{2}\mathbb{E}[\mathrm{sign}(\langle v, u' \rangle)u'] \\
&= \frac{1}{2}\mathbb{E}[\mathrm{sign}(\langle v, u \rangle)u] + \frac{1}{2}\mathbb{E}[\mathrm{sign}(\langle v, u \rangle)(2(\langle v, u \rangle)v - u)] \\
&= \mathbb{E}[(\langle v, u \rangle)\mathrm{sign}(\langle v, u \rangle)]v \\
&= \mathbb{E}[|\langle v, u \rangle|]v \\
&= \nu v,
\end{aligned}$$

where $\nu = \mathbb{E}[|\langle v, u \rangle|]$. The remaining thing is to derive lower and upper bounds for $\nu$.

We first derive an upper bound for $\nu$. There exists an orthogonal matrix $R \in O(d)$ with $Rv = e_d$ where $e_d$ is a vector with all zeros but the last element is 1, and set $w := Ru$. By symmetry of uniform distribution, $u' := R^{-1}u$, and $w$ follows $\mathrm{Unif}(\mathcal{S}_{\mathrm{T}_x\mathcal{M}}(1))$ as well. Thus we have $\mathbb{E}[|\langle v, u \rangle|] = \mathbb{E}[|\langle Rv, R^{-1}u \rangle|] = \mathbb{E}[|\langle e_1, u' \rangle|] = \mathbb{E}[|u_1'|]$. We observe that

$$\mathbb{E}[u_1^2] = \frac{1}{d}\mathbb{E}\left[\sum_{i=1}^d u_i^2\right] = \frac{1}{d},$$

and thus get the following upper bound for $\nu$:

$$\nu = \mathbb{E}[|\langle v, u \rangle|] = \mathbb{E}[|u_1'|] = \mathbb{E}[|u_1|] \leq \sqrt{\mathbb{E}[u_1^2]} = \frac{1}{\sqrt{d}}. \tag{17}$$

Next we derive a lower bound for $\nu$. Using $\langle v, u \rangle_x = \langle Rv, Ru \rangle_x = \langle e_d, w \rangle_x$, we have

$$\nu v = \mathbb{E}[\mathrm{sign}(\langle v, u \rangle_x)u] = \mathbb{E}[\mathrm{sign}(\langle Rv, Ru \rangle_x)R^\top(Ru)] = R^\top \mathbb{E}[\mathrm{sign}(\langle e_d, w \rangle_x)w]. \tag{18}$$

Observe that the term $\mathrm{sign}(\langle e_d, w \rangle_x)$ depends only on the $d$-th coordinate $w_d$. Due to the rotational symmetry of the uniform distribution of $w$, for any $k \neq d$, the expectation of the $k$-th component $\mathbb{E}[\mathrm{sign}(w_d)w_k]$ vanishes. So $\mathbb{E}[\mathrm{sign}(\langle e_d, w \rangle_x)w] = \mathbb{E}[\mathrm{sign}(w_d)w] = C_d e_d$ for some scalar $C_d > 0$. Therefore, from (18) we know that $\nu v = C_d R^\top e_d = C_d v$, which implies

$$\nu = C_d = \langle \mathbb{E}[\mathrm{sign}(w_d)w], e_d \rangle_x = \mathbb{E}[\mathrm{sign}(w_d)w_d] = \mathbb{E}[|w_d|].$$

Now, the marginal density of $w_d \in [-1, 1]$ is (Vershynin, 2009):

$$f(w_d) = A_d(1 - w_d^2)^{(d-3)/2} \text{ where } A_d = \frac{\Gamma\left(\frac{d}{2}\right)}{\sqrt{\pi}\Gamma\left(\frac{d-1}{2}\right)},$$

which yields

$$\nu = \mathbb{E}[|w_d|] = A_d \int_{-1}^1 |t|(1 - t^2)^{(d-3)/2}dt = 2A_d \int_0^1 t(1 - t^2)^{(d-3)/2}dt \geq \frac{1}{\sqrt{2\pi d}}, \tag{19}$$

where the last inequality follows the Gautschi's inequality.

Note that $\hat{C}$ in (6) satisfies $\hat{C} = \nu\sqrt{d}$, and therefore, by combining (17) and (19), we obtain $\hat{C} \in [1/\sqrt{2\pi}, 1]$, and this completes the proof. $\qquad\square$

### C.3. Proof of Proposition 3.3

*Proof.* For any vector $v \in \mathcal{S}_{T_x\mathcal{M}}(1)$, denote

$$A := \text{sign}(f(\text{Exp}_x(\nu u)) - f(\text{Exp}_x(-\nu u)))\langle u, v\rangle, \quad B := \text{sign}(\langle\text{grad}f(x), u\rangle_x)\langle u, v\rangle.$$

From Lemma 3.1 we know that $\mathbb{P}(A = B) \geq 1 - \gamma_x$, which together with $|A - B| \leq 2|\langle u, v\rangle| \leq 2$, yields $|\mathbb{E}_u A - \mathbb{E}_u B| \leq 2\gamma_x$. From Lemma 3.2, we know

$$\mathbb{E}_u B = \mathbb{E}_u\big[\text{sign}(\langle\text{grad}f(x), u\rangle_x)u\big] = \frac{\hat{C}}{\sqrt{d}}\frac{\text{grad}f(x)}{\|\text{grad}f(x)\|},$$

which completes the proof. $\qquad\square$

# D. Proof of Theorem 3.6

## D.1. Proof of Theorem 3.6 Part (i)

*Remark* D.1 (Expectation Notation). Unless specified otherwise by a subscript, the operator $\mathbb{E}[\cdot]$ denotes the total expectation taken over all random variables $\mathcal{U}_k$ generated up to the current iteration. We use the subscript notation (e.g., $\mathbb{E}_{u_k}[\cdot \mid x_k]$) for conditional expectations with respect to a specific random source $u_k$ given the history $x_k$.

*Proof.* For the ease of notation, we denote $h_k = h_\nu(x_k)$ when there is no confusion. Choosing $v = \frac{\text{grad}f(x_k)}{\|\text{grad}f(x_k)\|}$ in Proposition 3.3, we get

$$\mathbb{E}_{u_k}\big[\langle h_k, \text{grad}f(x_k)\rangle_{x_k} \mid x_k\big] \geq \frac{\hat{C}}{\sqrt{d}}\|\text{grad}f(x_k)\| - 2\gamma_{x_k}\|\text{grad}f(x_k)\|$$
$$\geq \frac{\hat{C}}{\sqrt{d}}\|\text{grad}f(x_k)\| - 2\sqrt{\frac{d}{2\pi}}L\nu. \tag{20}$$

Since $\mathcal{X} = \mathcal{M}$, we have $x_{k+1} = \mathcal{P}_\mathcal{X}\left(\text{Exp}_{x_k}(-\eta h_\nu(x_k))\right) = \text{Exp}_{x_k}(-\eta h_\nu(x_k))$. By Assumption 2.1, we get

$$f(x_{k+1}) \leq f(x_k) - \eta\langle h_k, \text{grad}f(x_k)\rangle_{x_k} + \frac{L\eta^2}{2},$$

which implies

$$\mathbb{E}_{u_k}\big[\langle h_k, \text{grad}f(x_k)\rangle_{x_k} \mid x_k\big] \leq \frac{1}{\eta}\mathbb{E}_{u_k}\big[f(x_k) - f(x_{k+1}) \mid x_k\big] + \frac{\eta L}{2}. \tag{21}$$

Combining (20) and (21) yields

$$\|\text{grad}f(x_k)\| \leq \frac{\sqrt{d}}{\hat{C}}\left(\frac{1}{\eta}\mathbb{E}_{u_k}\big[f(x_k) - f(x_{k+1}) \mid x_k\big] + \frac{\eta L}{2} + 2\sqrt{\frac{d}{2\pi}}L\nu\right). \tag{22}$$

Summing over $k = 0, 1, \ldots, T - 1$ and taking the total expectation on both sides, we apply the law of total expectation to obtain

$$
\begin{aligned}
\mathbb{E}\left[\sum_{k=0}^{T-1} \|\mathrm{grad}f(x_k)\|\right] &\leq \sum_{k=0}^{T-1} \frac{\sqrt{d}}{\hat{C}}\left(\frac{1}{\eta}\mathbb{E}\left[\mathbb{E}_{u_k}\left[f(x_k) - f(x_{k+1}) \,\big|\, x_k\right]\right] + \frac{\eta L}{2} + 2\sqrt{\frac{d}{2\pi}}L\nu\right) \\
&= \frac{\sqrt{d}}{\eta\hat{C}}\sum_{k=0}^{T-1}\mathbb{E}\left[f(x_k) - f(x_{k+1})\right] + \frac{T\sqrt{d}}{\hat{C}}\frac{\eta L}{2} + \frac{Td}{\hat{C}}2\sqrt{\frac{1}{2\pi}}L\nu \\
&= \frac{\sqrt{d}}{\eta\hat{C}}\mathbb{E}\left[f(x_0) - f(x_T)\right] + \frac{T\sqrt{d}}{\hat{C}}\frac{\eta L}{2} + \frac{Td}{\hat{C}}2\sqrt{\frac{1}{2\pi}}L\nu \\
&\leq \frac{\sqrt{d}}{\eta\hat{C}}(f(x_0) - f^*) + \frac{T\eta L\sqrt{d}}{2\hat{C}} + \frac{2TL\nu d}{\hat{C}\sqrt{2\pi}}.
\end{aligned}
\tag{23}
$$

Let $R$ be a random variable uniformly distributed over $\{0, \ldots, T - 1\}$, i.e., $\mathbb{P}(R = k) = 1/T$. The expected gradient norm at $x_R$ is given by

$$
\mathbb{E}[\|\mathrm{grad}f(x_R)\|] = \frac{1}{T}\sum_{k=0}^{T-1}\mathbb{E}[\|\mathrm{grad}f(x_k)\|].
$$

Dividing (23) by $T$ and using $\eta = \frac{1}{\sqrt{T}}$, we have

$$
\begin{aligned}
\mathbb{E}\left[\|\mathrm{grad}f(x_R)\|\right] &\leq \frac{1}{T}\frac{\sqrt{d}}{\eta\hat{C}}(f(x_0) - f^*) + \frac{\sqrt{d}}{\hat{C}}\frac{\eta L}{2} + \frac{d}{\hat{C}}2\sqrt{\frac{1}{2\pi}}L\nu \\
&\leq \frac{1}{T}\left(\frac{\sqrt{d}}{\eta\hat{C}}\frac{LD}{2}\right) + \frac{\sqrt{d}}{\hat{C}}\frac{\eta L}{2} + \frac{d}{\hat{C}}2\sqrt{\frac{1}{2\pi}}L\nu \\
&= \frac{1}{\sqrt{T}}\frac{\sqrt{d}}{\hat{C}}\frac{LD}{2} + \frac{1}{\sqrt{T}}\frac{\sqrt{d}}{\hat{C}}\frac{L}{2} + \frac{d}{\hat{C}}2\sqrt{\frac{1}{2\pi}}L\nu \\
&= \frac{1}{2\sqrt{T}}\left(\frac{L\sqrt{d}}{\hat{C}}(D + 1)\right) + \frac{2dL}{\hat{C}\sqrt{2\pi}}\nu.
\end{aligned}
$$

Choosing $T$ and $\nu$ as specified in Theorem 3.6 Part (i) yields the desired result. $\qquad\square$

## D.2. Proof of Theorem 3.6 Part (ii)

*Proof.* The derivation is analogous to the previous proof, with the only difference being the choice of step size. By simply substituting the cosine annealing step size $\eta_k$ for $\eta$ in (20), (21), and (22), and summing over $k = 0, \ldots, T - 1$, the argument proceeds identically. Following similar argument as (23), we have

$$
\frac{\hat{C}}{\sqrt{d}}\sum_{k=0}^{T-1}\eta_k\mathbb{E}\left[\|\mathrm{grad}f(x_k)\|\right] \leq f(x_0) - f^* + \frac{L}{2}\sum_{k=0}^{T-1}\eta_k^2 + 2\sqrt{\frac{d}{2\pi}}L\nu\sum_{k=0}^{T-1}\eta_k.
$$

We define a random variable $R$ taking values in $\{0, \ldots, T - 1\}$ with probability $\mathbb{P}(R = k) = \frac{\eta_k}{\sum_{j=0}^{T-1}\eta_j}$. Then, the left-hand side can be rewritten as

$$
\frac{\hat{C}}{\sqrt{d}}\left(\sum_{k=0}^{T-1}\eta_k\right)\mathbb{E}[\|\mathrm{grad}f(x_R)\|].
$$

Dividing both sides by $\frac{\hat{C}}{\sqrt{d}}\sum_{k=0}^{T-1}\eta_k$ and using Lemma B.1, we obtain

$$
\begin{aligned}
\mathbb{E}\left[\|\mathrm{grad}f(x_R)\|\right] &\leq \frac{\sqrt{d}}{\hat{C}}\left[\frac{LD}{\frac{\eta_0}{2}(T+1)} + \frac{L}{2}\cdot\frac{\frac{\eta_0^2}{8}(3T+4)}{\frac{\eta_0}{2}(T+1)} + 2\sqrt{\frac{d}{2\pi}}L\nu\right] \\
&= \frac{\sqrt{d}}{\hat{C}}\left[\frac{2LD}{\eta_0(T+1)} + \frac{L\eta_0}{8}\cdot\frac{3T+4}{T+1} + 2\sqrt{\frac{d}{2\pi}}L\nu\right]
\end{aligned}
$$

$$\leq \frac{\sqrt{d}}{\hat{C}} \left[ \frac{2LD}{\eta_0(T+1)} + \frac{4L\eta_0}{8} + 2\sqrt{\frac{d}{2\pi}}L\nu \right]$$

$$\leq \frac{\sqrt{d}}{\hat{C}} \left[ \frac{2LD}{\sqrt{T}} + \frac{L}{2\sqrt{T}} + 2\sqrt{\frac{d}{2\pi}}L\nu \right].$$

Choosing $T$ and $\nu$ as in (8) yields the desired result. $\qquad\square$

## E. Proof of Theorem 3.7

### E.1. Proof of Theorem 3.7 Part (i)

*Proof.* We prove by contradiction. Note that the algorithm returns $\hat{x}_T$, and $f(\hat{x}_T) = \min_{0 \leq k \leq T} f(x_k)$. Suppose that $f(\hat{x}_T) \geq f(x^*)+\epsilon$. This implies $f(x_k) \geq f(x^*)+\epsilon$ for all $k \in \{0,\ldots,T-1\}$. We have $\bar{\zeta} \geq \zeta(\kappa, \bar{d}(x_k, x^*))$ for all $x_k \in \mathcal{X}$, as $\zeta$ is increasing in the second entry and $\mathfrak{D}$ is the diameter of $\mathcal{X}$. By Lemma B.3, we get $d(x_k, x^*) \leq \sqrt{D}+\eta k \leq \sqrt{D}+\eta T$ for $k < T$. By Lemma B.4, we get $\|\mathrm{grad} f(x_k)\| \geq \frac{\epsilon}{d(x_k,x^*)}$. Thus, $\nu$ defined in (9) satisfies

$$\nu = \frac{1}{4\sqrt{2}d} \frac{(\epsilon/L)^{3/2}}{(\sqrt{D}+\eta T)^2} \leq \frac{1}{4\sqrt{2}d} \frac{(\epsilon/L)^{3/2}}{d(x_k,x^*)^2} \leq \frac{1}{4\sqrt{2}Ld} \frac{\|\mathrm{grad} f(x_k)\|}{d(x_k,x^*)} \sqrt{\frac{\epsilon}{L}}, \forall k \in [T]. \qquad (24)$$

From Lemma B.2 and Proposition 3.3, we have the recursive bound:

$$\mathbb{E}_{u_k}\left[d^2(x_{k+1},x^*)|x_k\right] \leq d^2(x_k,x^*) - \frac{1}{\sqrt{\pi}}\frac{\sqrt{\epsilon}}{\sqrt{dL}}\eta + \left(\sqrt{\frac{8d}{\pi}}\frac{L\|\mathrm{Log}_{x_k}(x^*)\|}{\|\mathrm{grad} f(x_k)\|}\nu\right)\eta + \bar{\zeta}\eta^2$$

$$\leq d^2(x_k,x^*) - \frac{1}{\sqrt{\pi}}\frac{\sqrt{\epsilon}}{\sqrt{dL}}\eta + \frac{1}{2\sqrt{\pi}}\frac{\sqrt{\epsilon}}{\sqrt{dL}}\eta + \bar{\zeta}\eta^2$$

$$= d^2(x_k,x^*) - \frac{\epsilon}{16\pi dL\bar{\zeta}}, \qquad (25)$$

where the equality is due to $\eta$ defined in (9).

Taking the full expectation on both sides of (25) and iterating the inequality recursively for $k = 0,\ldots,T-1$, we derive

$$0 \leq \mathbb{E}\left[d^2(x_T,x^*)\right] \leq d^2(x_0,x^*) - \frac{\epsilon}{16\pi dL\bar{\zeta}}T < 0,$$

where the last inequality is due to the choice of $T$ in (9). This is a contradiction. Thus, there must exist an iteration $k$ such that $f(x_k) - f(x^*) < \epsilon$, which implies $f(\hat{x}_T) - f(x^*) < \epsilon$. $\qquad\square$

### E.2. Proof of Theorem 3.7 Part (ii)

*Proof.* We follow the same contradiction argument as in Part (i). Suppose that $f(x_k) \geq f(x^*)+\epsilon$ for all $k \in \{0,\ldots,T-1\}$. Using the cosine annealing step size $\eta_k$, the accumulated step length is $\sum_{j=0}^{T-1}\eta_j = \frac{\eta_0}{2}(T+1)$. For the choice of $\nu$ in (10), we use the bound involving $T$ as a sufficient approximation for large $T$, ensuring (24) holds.

Replacing $\eta$ with $\eta_k$ in the derivation of (25), we obtain:

$$\mathbb{E}_{u_k}\left[d^2(x_{k+1},x^*)|x_k\right] \leq d^2(x_k,x^*) - \frac{1}{\sqrt{\pi}}\frac{\sqrt{\epsilon}}{\sqrt{dL}}\eta_k + \frac{1}{2\sqrt{\pi}}\frac{\sqrt{\epsilon}}{\sqrt{dL}}\eta_k + \bar{\zeta}\eta_k^2$$

$$= d^2(x_k,x^*) - \frac{\sqrt{\epsilon}}{2\sqrt{\pi dL}}\eta_k + \bar{\zeta}\eta_k^2.$$

Taking the full expectation and summing the inequality over $k = 0,\ldots,T-1$, we get

$$\mathbb{E}\left[d^2(x_T,x^*)\right] \leq d^2(x_0,x^*) - \frac{\sqrt{\epsilon}}{2\sqrt{\pi dL}}\sum_{k=0}^{T-1}\eta_k + \bar{\zeta}\sum_{k=0}^{T-1}\eta_k^2.$$

By Lemma B.1, we have $\sum_{k=0}^{T-1} \eta_k = \frac{\eta_0}{2}(T+1)$ and $\sum_{k=0}^{T-1} \eta_k^2 = \frac{\eta_0^2}{8}(3T+4)$. Using the bound $\sum_{k=0}^{T-1} \eta_k^2 \le \eta_0 \sum_{k=0}^{T-1} \eta_k$ (since $\frac{3T+4}{T+1} \le 4$), we have:

$$\mathbb{E}\left[d^2(x_T, x^*)\right] \le D - \left(\frac{\sqrt{\epsilon}}{2\sqrt{\pi dL}} - \bar{\zeta}\eta_0\right) \sum_{k=0}^{T-1} \eta_k$$

$$= D - \frac{\sqrt{\epsilon}}{4\sqrt{\pi dL}} \sum_{k=0}^{T-1} \eta_k$$

$$= D - \frac{\sqrt{\epsilon}}{4\sqrt{\pi dL}} \frac{\eta_0}{2}(T+1).$$

Substituting $\eta_0 = \frac{\sqrt{\epsilon}}{4\sqrt{\pi}\bar{\zeta}\sqrt{dL}}$:

$$\mathbb{E}\left[d^2(x_T, x^*)\right] < D - \frac{\sqrt{\epsilon}}{8\sqrt{\pi dL}} \left(\frac{\sqrt{\epsilon}}{4\sqrt{\pi}\bar{\zeta}\sqrt{dL}}\right) T$$

$$= D - \frac{\epsilon}{32\pi dL\bar{\zeta}}T.$$

Given $T \ge 1 + \frac{32\pi dL\bar{\zeta}}{\epsilon}D$, we have $\mathbb{E}\left[d^2(x_T, x^*)\right] < 0$, which is a contradiction. $\qquad\square$

## F. Proof of Theorem 3.10

*Proof.* Note that in each iteration of Algorithm 2, we run Algorithm 1 for $t = \lceil 1 + 64\pi dL\bar{\zeta}/\alpha \rceil$ iterations. From Theorem 3.7 (i), with the choices of $\eta_k$ and $\nu_k$ defined in Theorem 3.10, we have that for any phase $k \in \{0, \ldots, K-1\}$, starting from $x^{(k)}$ with initial distance bound $D_k \ge \mathbb{E}[d^2(x^{(k)}, x^*)]$:

$$\mathbb{E}[f(x^{(k+1)})] - f(x^*) \le \epsilon_k = \frac{\alpha D_k}{4},$$

with oracle complexity $2t$. Since $f$ is geodesically $\alpha$-strongly convex, we have

$$\mathbb{E}\left[\frac{\alpha}{2}d^2(x^{(k+1)}, x^*)\right] \le \mathbb{E}[f(x^{(k+1)})] - f(x^*) \le \frac{\alpha D_k}{4},$$

which implies

$$\mathbb{E}[d^2(x^{(k+1)}, x^*)] \le \frac{D_k}{2} = D_{k+1}.$$

This justifies the update rule in the algorithm. After $K$ phases, the expected distance squared of the final output $x^{(K)}$ is bounded by:

$$\mathbb{E}[d^2(x^{(K)}, x^*)] \le \frac{D}{2^K}.$$

Note that Assumption 2.1 and the fact that $\mathcal{X}$ is a bounded set imply that $f$ is geodesically Lipschitz continuous with some Lipschitz constant $l$. This further implies

$$\mathbb{E}[f(x^{(K)}) - f(x^*)] \le l \cdot \mathbb{E}[d(x^{(K)}, x^*)] \le l\sqrt{\mathbb{E}[d^2(x^{(K)}, x^*)]} \le l\sqrt{\frac{D}{2^K}} \le \epsilon,$$

where the second inequality is Jensen's inequality, and the last inequality is due to the definition of $K$. $\qquad\square$

*Remark* F.1. All results developed in Section 3 use the search direction $h_\nu$ defined in (4), which is based on a single pair of samples. It is worth noting that all results in Section 3 also hold for the batch averaged estimator $\bar{h}$ defined in (12). This is because our analysis relies solely on the expectation of $h_\nu$, which is identical to the expectation of $\bar{h}$. For more details, please refer to (27).

# G. Proof of Theorem 4.1

*Proof.* We denote $\mathfrak{M}_f := L\mathfrak{D}^2$, $\tilde{C} = \sqrt{d}/\hat{C}$, normalized Riemannian gradient $w_k := \frac{\operatorname{grad} f(x_k)}{\|\operatorname{grad} f(x_k)\|}$, the function value gap by $\Delta_k := f(x_k) - f(x^*)$ for $k = 0, 1, \ldots, T$. Let $\mathcal{B}_k := \{u_1^{(k)}, \ldots, u_{M_k}^{(k)}\}$ denote the batch of i.i.d. random directions sampled at iteration $k$.

By geodesically $L$-smoothness, we have

$$
\begin{aligned}
\Delta_{k+1} &= f(x_{k+1}) - f(x^*) \\
&\leq f(x_k) - f(x^*) + s_k\|\operatorname{grad} f(x_k)\|\langle w_k, \operatorname{Log}_{x_k}(z_k)\rangle + \tfrac{1}{2}s_k^2\mathfrak{M}_f \\
&= \Delta_k + s_k\|\operatorname{grad} f(x_k)\|\langle \tilde{C}\bar{h}_k, \operatorname{Log}_{x_k}(z_k)\rangle + s_k\|\operatorname{grad} f(x_k)\|\langle w_k - \tilde{C}\bar{h}_k, \operatorname{Log}_{x_k}(z_k)\rangle + \tfrac{1}{2}s_k^2\mathfrak{M}_f \\
&\leq \Delta_k + s_k\|\operatorname{grad} f(x_k)\|\langle \tilde{C}\bar{h}_k, \operatorname{Log}_{x_k}(x^*)\rangle + s_k\|\operatorname{grad} f(x_k)\|\langle w_k - \tilde{C}\bar{h}_k, \operatorname{Log}_{x_k}(z_k)\rangle + \tfrac{1}{2}s_k^2\mathfrak{M}_f \\
&= \Delta_k + s_k\|\operatorname{grad} f(x_k)\|\langle w_k, \operatorname{Log}_{x_k}(x^*)\rangle + \tfrac{1}{2}s_k^2\mathfrak{M}_f + s_k\|\operatorname{grad} f(x_k)\|\langle w_k - \tilde{C}\bar{h}_k, \operatorname{Log}_{x_k}(z_k) - \operatorname{Log}_{x_k}(x^*)\rangle,
\end{aligned}
$$

where the second inequality is due to the definition of $z_k$ in Algorithm 3. Taking expectation on both sides, we get

$$
\begin{aligned}
&\mathbb{E}_{\mathcal{B}_k}[\Delta_{k+1} \mid x_k] \\
&\leq \Delta_k + s_k\langle \operatorname{grad} f(x_k), \operatorname{Log}_{x_k}(x^*)\rangle + \tfrac{1}{2}s_k^2\mathfrak{M}_f + s_k\|\operatorname{grad} f(x_k)\|\mathbb{E}_{\mathcal{B}_k}[\langle w_k - \tilde{C}\bar{h}_k, \operatorname{Log}_{x_k}(z_k) - \operatorname{Log}_{x_k}(x^*)\rangle \mid x_k] \\
&\leq \Delta_k - s_k\Delta_k + \tfrac{1}{2}s_k^2\mathfrak{M}_f + 2\mathfrak{D}s_k\|\operatorname{grad} f(x_k)\| \cdot \mathbb{E}_{\mathcal{B}_k}[\|w_k - \tilde{C}\bar{h}_k\| \mid x_k],
\end{aligned}
\tag{26}
$$

where the last inequality used the geodesic convexity property (3). Next we derive an upper bound for $\mathbb{E}_{\mathcal{B}_k}[\|w_k - \tilde{C}\bar{h}_k\| \mid x_k]$. We first show that the expectation of the batch-average estimator $\bar{h}_k$ is the same as the expectation of the single sample estimator $h_k := h_{\nu_k}(x_k)$ defined in (4). Denoting $\bar{m}_k := \mathbb{E}_{\mathcal{B}_k}[\bar{h}_k \mid x_k]$ and $m_k := \mathbb{E}_{u_k}[h_k \mid x_k]$, we have

$$
\bar{m}_k = \mathbb{E}_{\mathcal{B}_k}[\bar{h}_k \mid x_k] = \frac{1}{M_k}\sum_{j=1}^{M_k}\mathbb{E}_{u_j^{(k)}}[o_j^{(k)}u_j^{(k)} \mid x_k] = \mathbb{E}_{u_k}[o_k u_k \mid x_k] = \mathbb{E}_{u_k}[h_k \mid x_k] = m_k.
\tag{27}
$$

Next we bound the variance. By independence of $\{u_j\}_j$ and $\mathbb{E}_{u_j^{(k)}}[o_j^{(k)}u_j^{(k)}|x_k] - m_k = 0$, we have

$$
\mathbb{E}_{\mathcal{B}_k}[\|\bar{h}_k - \bar{m}_k\|_2^2 \mid x_k] = \frac{1}{M_k^2}\sum_{j=1}^{M_k}\mathbb{E}_{u_j^{(k)}}[\|o_j^{(k)}u_j^{(k)} - m_k\|_2^2 \mid x_k]
$$

$$
= \frac{1}{M_k^2}\sum_{j=1}^{M_k}(\mathbb{E}_{u_j^{(k)}}[\|o_j^{(k)}u_j^{(k)}\|_2^2 \mid x_k] - \|m_k\|_2^2) = \frac{1}{M_k}(1 - \|m_k\|^2).
$$

By Jensen's inequality, we get

$$
\mathbb{E}_{\mathcal{B}_k}\left[\|\bar{h}_k - \bar{m}_k\| \mid x_k\right] \leq \sqrt{\mathbb{E}_{\mathcal{B}_k}\left[\|\bar{h}_k - \bar{m}_k\|^2 \mid x_k\right]} = \sqrt{(1 - \|m_k\|^2)/M_k} \leq 1/\sqrt{M_k}.
\tag{28}
$$

As a consequence of the above inequalities, we get that

$$
\begin{aligned}
\mathbb{E}_{\mathcal{B}_k}[\|w_k - \tilde{C}\bar{h}_k\| \mid x_k] &= \mathbb{E}_{\mathcal{B}_k}[\|(w_k - \tilde{C}\bar{m}_k) + \tilde{C}(\bar{m}_k - \bar{h}_k)\| \mid x_k] \\
&\leq \|w_k - \tilde{C}\bar{m}_k\| + \tilde{C}\mathbb{E}_{\mathcal{B}_k}[\|\bar{h}_k - \bar{m}_k\| \mid x_k] \\
&= \tilde{C}\|w_k/\tilde{C} - \bar{m}_k\| + \tilde{C}\mathbb{E}_{\mathcal{B}_k}[\|\bar{h}_k - \bar{m}_k\| \mid x_k] \\
&\leq 2\tilde{C}\gamma_{x_k} + \tilde{C}/\sqrt{M_k},
\end{aligned}
\tag{29}
$$

where the last inequality is from (28) and Proposition 3.3 by setting $v = \frac{\bar{m}_k - w_k/\tilde{C}}{\|\bar{m}_k - w_k/\tilde{C}\|}$.

Combining (26) and (29), we get

$$
\mathbb{E}_{\mathcal{B}_k}\left[\Delta_{k+1}|x_k\right]
$$

$$\leq \Delta_k - s_k \Delta_k + \tfrac{1}{2} s_k^2 \mathfrak{M}_f + 2\mathfrak{D} s_k \|\operatorname{grad} f(x_k)\| (2\tilde{C}\gamma_{x_k} + \tilde{C}/\sqrt{M_k})$$

$$= \Delta_k - s_k \Delta_k + \tfrac{1}{2} s_k^2 \mathfrak{M}_f + 2\mathfrak{D} s_k \|\operatorname{grad} f(x_k)\| \left( \tilde{C}\sqrt{\frac{2d}{\pi}} \frac{L\nu_k}{\|\operatorname{grad} f(x_k)\|} + \frac{\tilde{C}}{\sqrt{M_k}} \right)$$

$$\leq \Delta_k - s_k \Delta_k + \tfrac{1}{2} s_k^2 \mathfrak{M}_f + 2\mathfrak{D} s_k \left( \tilde{C}\sqrt{\frac{2d}{\pi}} L\nu_k + \frac{\sqrt{2L\Delta_k}\tilde{C}}{\sqrt{M_k}} \right)$$

$$\leq \Delta_k - s_k \Delta_k + \tfrac{1}{2} s_k^2 \mathfrak{M}_f + 2\mathfrak{D} s_k \tilde{C}\sqrt{\frac{2d}{\pi}} L\nu_k + \left( \frac{\Delta_k}{4} + \frac{8L\mathfrak{D}^2 d}{\hat{C}^2 M_k} \right) s_k$$

$$= \left( 1 - \frac{3 s_k}{4} \right) \Delta_k + \frac{s_k^2}{2} (\mathfrak{M}_f + 3),$$

where the second inequality is from Lemma B.6, the third inequality used Young's inequality, and the last equality is obtained by substituting the choices of $s_k$ in Algorithm 3 and $\nu_k$ and $M_k$ in Theorem 4.1. Denote $a_k := 1 - \frac{3}{4} s_k = 1 - \frac{3/2}{k+3}$. Taking the total expectation on both sides of the above inequality and applying it recursively, we obtain

$$\mathbb{E}[\Delta_k] \leq \left( \prod_{t=0}^{k-1} a_t \right) \Delta_0 + \sum_{j=0}^{k-1} \frac{2(\mathfrak{M}_f + 3)}{(j+3)^2} \prod_{t=j+1}^{k-1} a_t. \tag{30}$$

We now develop upper bounds for the two terms on the right hand side of (30). Note that

$$a_k = 1 - \frac{3}{2} \cdot \frac{1}{k+3} \leq \left( 1 - \frac{1}{k+3} \right)^{3/2} = \left( \frac{k+2}{k+3} \right)^{3/2}.$$

Therefore,

$$\prod_{t=0}^{k-1} a_t \leq \left( \prod_{t=0}^{k-1} \frac{t+2}{t+3} \right)^{3/2} = \left( \frac{2}{k+2} \right)^{3/2} \leq \frac{2}{k+2}, \quad \text{and} \quad \prod_{t=j+1}^{k-1} a_t \leq \left( \prod_{t=j+1}^{k-1} \frac{t+2}{t+3} \right)^{3/2} = \left( \frac{j+3}{k+2} \right)^{3/2},$$

which further implies

$$\sum_{j=0}^{k-1} \frac{2(\mathfrak{M}_f + 3)}{(j+3)^2} \prod_{t=j+1}^{k-1} a_t \leq \sum_{j=0}^{k-1} \frac{2(\mathfrak{M}_f + 3)}{(j+3)^2} \left( \frac{j+3}{k+2} \right)^{\frac{3}{2}} = \frac{2(\mathfrak{M}_f + 3)}{(k+2)^{3/2}} \sum_{j=0}^{k-1} \frac{1}{\sqrt{j+3}} \leq \frac{4(\mathfrak{M}_f + 3)}{k+2},$$

where the last inequality is from that $\sum_{m=3}^{k+2} \frac{1}{\sqrt{m}} \leq \int_2^{k+2} x^{-1/2} dx \leq 2\sqrt{k+2}$.

Substituting these upper bounds to (30), we obtain

$$\mathbb{E}[\Delta_k] \leq \frac{2\Delta_0 + 4\mathfrak{M}_f + 12}{k+2}.$$

So it takes $T = \lceil \frac{2\Delta_0 + 4\mathfrak{M}_f + 12}{\epsilon} \rceil$ to achieve $\mathbb{E}[\Delta_T] \leq \epsilon$. $\qquad \square$

## H. Sensitivity of Frank-Wolfe Search Direction

A key difference between projected gradient methods and Frank–Wolfe type methods is how they respond to noise in the gradient direction estimate. This effect is visible even in Euclidean case. In projected normalized gradient descent, the update takes the following form with noisy gradient direction $\tilde{g}_t$: $x_{t+1} = \mathcal{P}_{\mathcal{X}}(x_t - \eta \tilde{g}_t)$, Due to non-expansiveness (Assumption 2.6) of the projection operator, the distance between $x_{t+1}$, and the hypothetical update that one would have gotten with the true normalized gradient $g_t$, is upper bounded by $\eta \|\tilde{g}_t - g_t\|$).

In contrast, Frank–Wolfe computes a search point via a linear minimization oracle

$$s(\tilde{g}_t) \in \arg\min_{s \in \mathcal{X}} \langle \tilde{g}_t, s - x \rangle,$$

where $\mathcal{X}$ is a compact convex set. For such problems, the solution $s(\tilde{g}_t)$ lies on the boundary of $\mathcal{X}$. As a result, even an arbitrarily small perturbation of the direction $\tilde{g}_t$ can cause the oracle to select a completely different boundary point. In other words, even a small difference between $g_t$ and $\tilde{g}_t$, can lead to completely different search directions $s(g_t)$ and $s(\tilde{g}_t)$ and $\|s(g_t) - s(\tilde{g}_t)\|$ can be in the order of diameter $\mathcal{O}(\mathfrak{D})$ (see Figure 6). This means the variance of $s(\tilde{g}_t)$ is of constant order. Therefore, to guarantee descent and convergence for Frank–Wolfe with noisy or comparison-based oracles, we must explicitly control the variance of the direction estimator, whereas projected gradient methods are much more stable under the same level of noise.

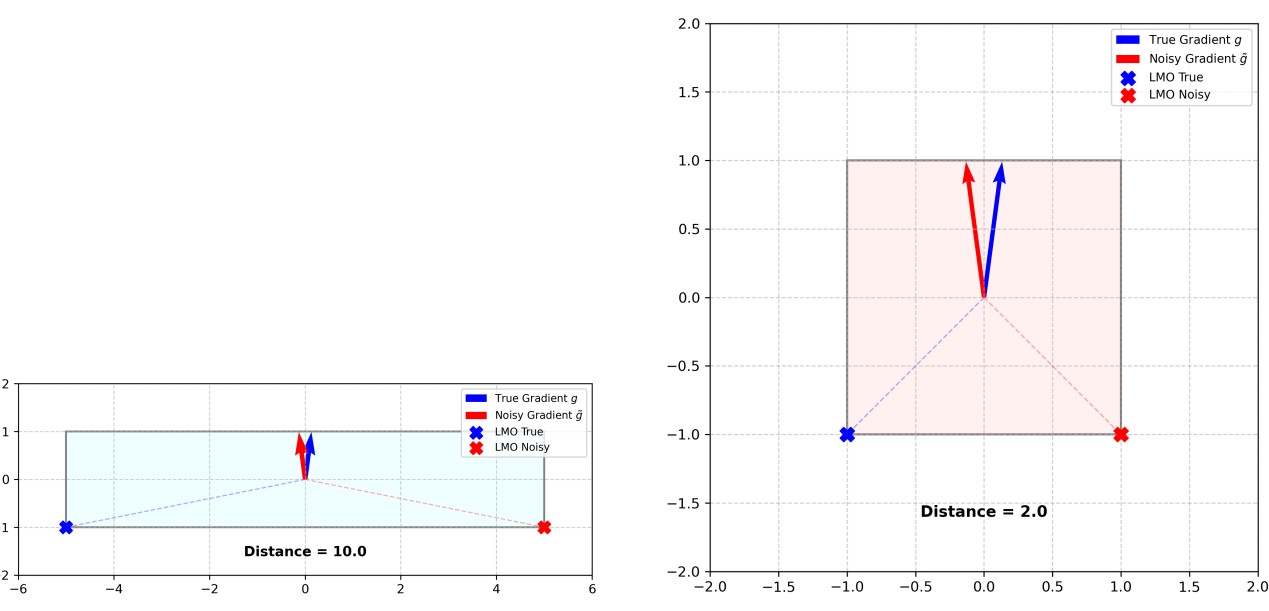

*(a)* Large diameter: Significant deviation between LMO solutions.    *(b)* Small diameter: Reduced deviation with constant noise.

*Figure 6.* Impact of the constraint set diameter on the sensitivity of the Frank-Wolfe Linear Minimization Oracle (LMO) to gradient noise. (a) A larger diameter leads to significant deviation between the true and noisy LMO solutions. (b) A smaller diameter mitigates this deviation, even when the noise magnitude remains constant. Overall, this demonstrates how gradient noise error is amplified by the diameter of the constraint set.

# I. Additional Numerical Experiments

## I.1. Additional Synthetic problem results

We provide the CPU time results for the Rayleigh quotient problem in Figure 7 and the Karcher Mean problem in Figure 8. In these low-dimensional synthetic settings, it can be observed that RDNGD converges more slowly than ZO-RGD. This occurs because RDNGD relies solely on a comparison oracle and lacks access to gradient magnitude information. Consequently, with an unit length search direction, RDNGD requires conservative step sizes to avoid oscillation around the minimum, whereas ZO-RGD benefits from gradient magnitude estimation for more efficient updates. This limitation is inherent to optimization problems utilizing only comparison oracles. However, for the high-dimensional real-world applications shown in Figure 4 and Figures 9–11, we observe a clear advantage in CPU time for RDNGD, demonstrating the efficiency of our method.

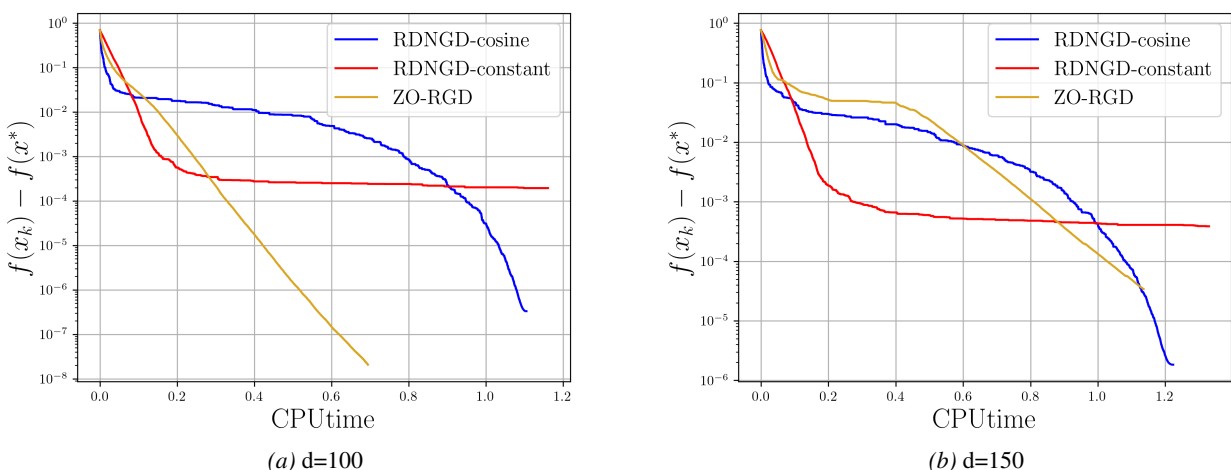

*(a)* d=100          *(b)* d=150

*Figure 7.* CPUtime results for Rayleigh quotient maximization.

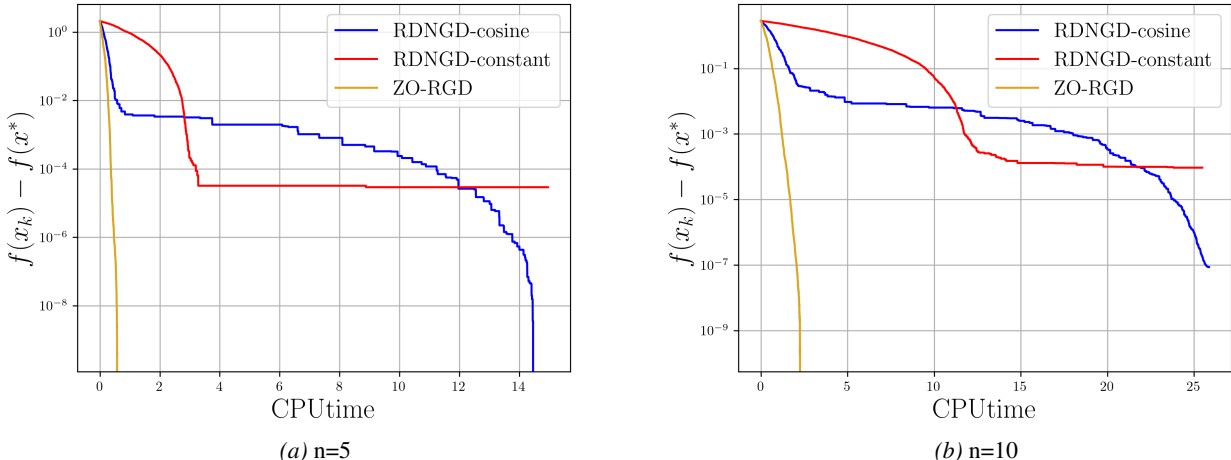

*(a)* n=5          *(b)* n=10

*Figure 8.* CPUtime result for Karcher mean problem.

## I.2. Additional Real Application Examples

In this section, we provide additional numerical results to supplement the main paper. We present more examples of the DNN attack applications in Figures 9–11, and additional results for the Horizon leveling task in Figures 12–14.

It is important to note that the experimental settings, model architectures, and hyperparameters used in these additional experiments are identical to those described in the main paper.

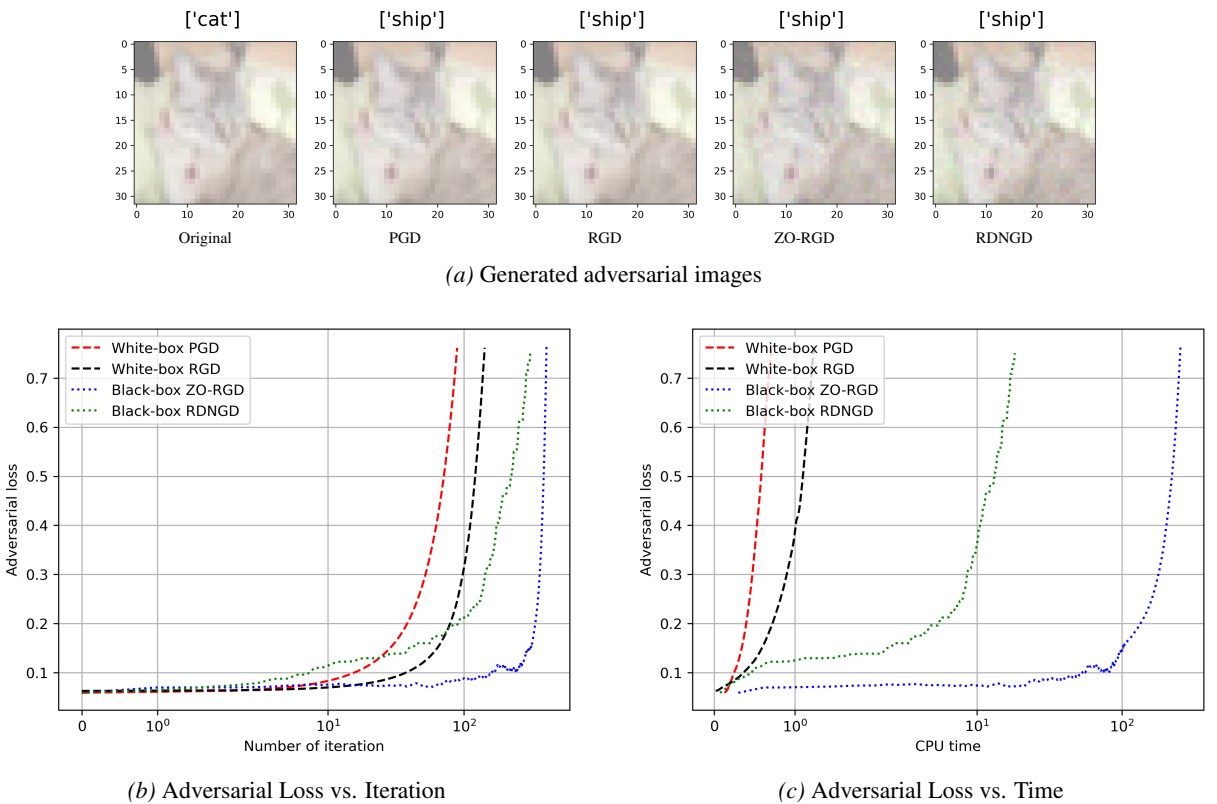

*(a)* Generated adversarial images

*(b)* Adversarial Loss vs. Iteration    *(c)* Adversarial Loss vs. Time

*Figure 9.* Additional attack result on CIFAR-10 (Example 1).

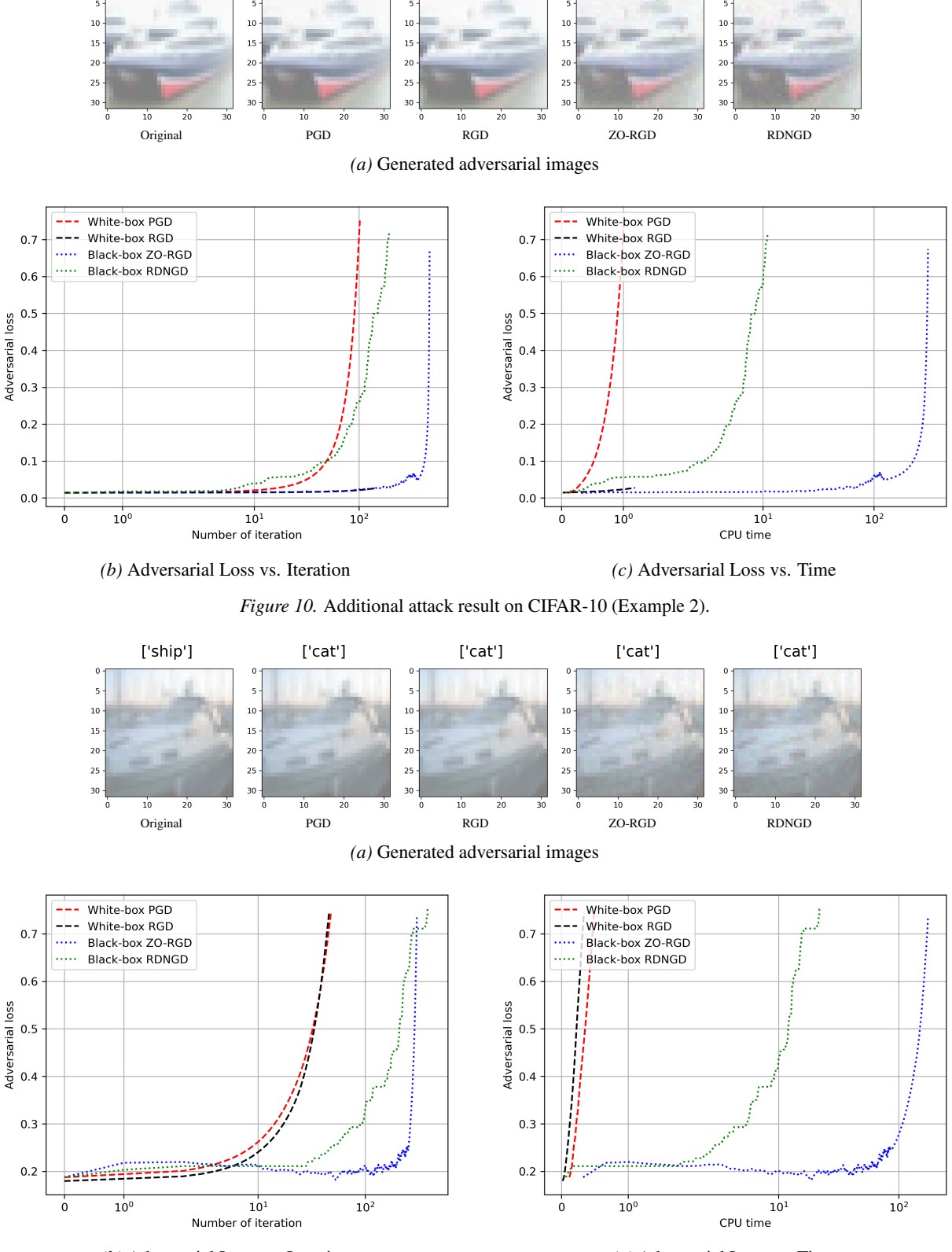

*(a)* Generated adversarial images

*(b)* Adversarial Loss vs. Iteration

*(c)* Adversarial Loss vs. Time

*Figure 10.* Additional attack result on CIFAR-10 (Example 2).

*(a)* Generated adversarial images

*(b)* Adversarial Loss vs. Iteration

*(c)* Adversarial Loss vs. Time

*Figure 11.* Additional attack result on CIFAR-10 (Example 3).

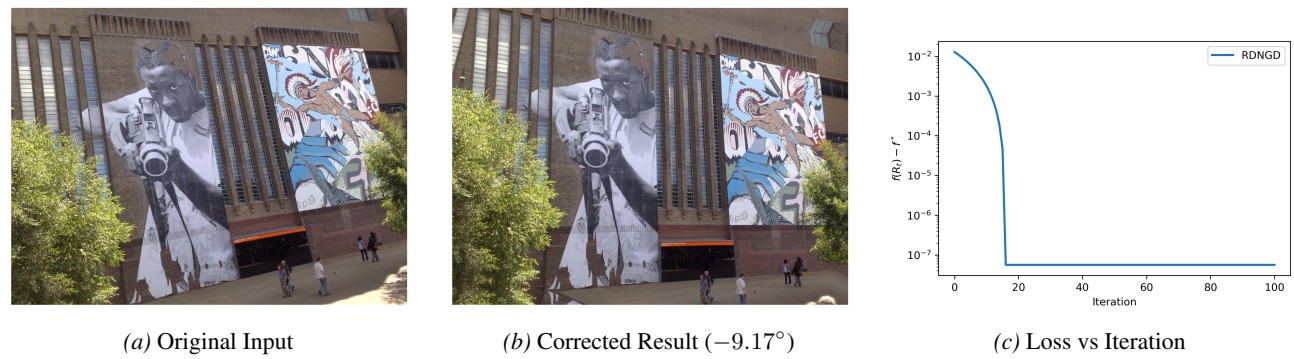

*(a)* Original Input     *(b)* Corrected Result $(-9.17°)$     *(c)* Loss vs Iteration

*Figure 12.* Additional horizon leveling result (Example 1).

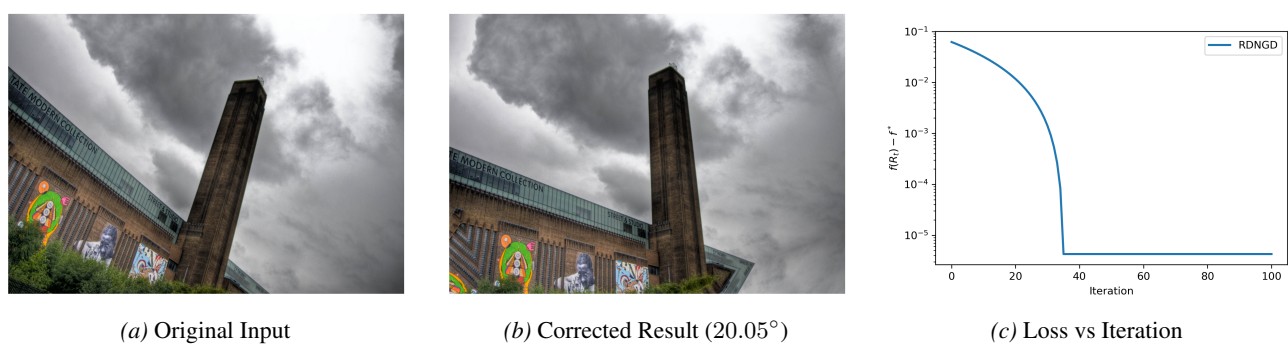

*(a)* Original Input     *(b)* Corrected Result $(20.05°)$     *(c)* Loss vs Iteration

*Figure 13.* Additional horizon leveling result (Example 2).

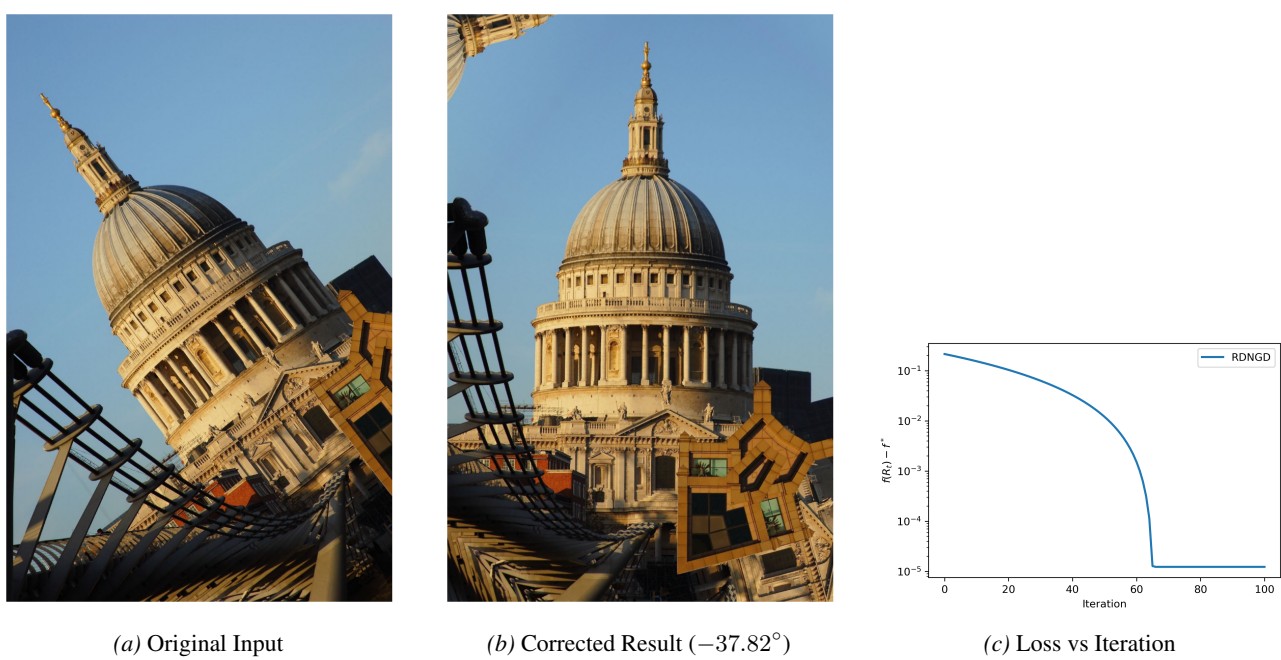

*(a)* Original Input     *(b)* Corrected Result $(-37.82°)$     *(c)* Loss vs Iteration

*Figure 14.* Additional horizon leveling result (Example 3).

