# OpenReview forum: "Riemannian Dueling Optimization"
_ICML.cc/2026/Conference — ICML 2026 regular_

### Official Review · Reviewer_AM8Q · 2026-03-04

**Soundness:** 3
**Presentation:** 3
**Significance:** 3
**Originality:** 3
**Overall Recommendation:** 5
**Confidence:** 3

**Summary:**

This paper focuses on the problem of dueling optimization under manifold constraints. As motivation, the authors present 'Attacks on Deep Neural Networks' and 'horizon leveling' as two practical examples where Riemannian dueling optimization is applicable.

**Compliance With Llm Reviewing Policy:**

Affirmed.

**Final Justification:**

This paper studies the problem of dueling optimization on Riemannian manifolds. The topic is novel, the motivation is well presented, the experiments are thorough, and the theoretical results are complete, making this a solid piece of research. However, the proposed methods require the batch size to grow with the number of iterations, and the choice of step size in the experiments is not theoretically justified, which may raise concerns about generalization. Nevertheless, considering the overall quality of the work, I recommend acceptance.

**Key Questions For Authors:**

see strength and weaknesses

**Limitations:**

yes

**Strengths And Weaknesses:**

As for the contributions, the authors employ RDNGD/RDFW methods for Riemannian dueling optimization and derive the oracle complexities for geodesically L-smooth/convex objectives. The overall oracle complexity of the four methods proposed in this paper is either lower than or comparable to existing NGD methods. The authors conducted experiments on both synthetic and real-world problems, demonstrating that the proposed methods achieve similar or even better performance than ZO-RGD while relying solely on comparison oracles.The mathematical proofs in this paper are rigorous. I appreciate the overall quality of this paper, but I still have some questions for the authors:

**Q1**  For which types of manifolds does the property of being 'geodesically uniquely convex' hold? In my understanding, this property does not hold for the sphere.

**Q2** Regarding Line 3 of Algorithm 1, uniform random sampling on $S_{T_{x_k}M}(1)$ can become complex when applied to other manifolds, such as constant-rank manifolds. Is there a simpler method to replace this step?

**Q3** Regarding the application scenarios for the RDNGD method, I am not entirely clear on the definition of a 'constrained convex problem where $f$ is geodesically convex and geodesically L-smooth.' Are there any concrete application examples of this in machine learning?

---

> ### Author Rebuttal · Authors · 2026-03-30
>
> We thank the reviewer for the insightful questions, comments, and suggestions.
>
> >Q1. For which types of manifolds does the property of being 'geodesically uniquely convex' hold? In my understanding, this property does not hold for the sphere.
>
> **Geodesic (unique) convexity is a property of the objective over a domain on the manifold, rather than of the manifold alone. There are many important examples in the literature.**
> 1. The Karcher (Fréchet) mean objective (Section 5.1.2) on SPD manifold [A8].
> 2. The $\log\det(X)$ function on the SPD manifold [A4].
> 3. Computing the optimal constant in the Brascamp-Lieb inequality [A5].
> 4. Regularized covariance/scatter estimation problems on SPD manifolds are geodesically convex [A6].
> 5. On the positive orthant equipped with a logarithmic metric, polynomials with positive coefficients and their logarithms are geodesically convex [A4].
> 6. Fréchet mean problem in hyperbolic space [A8].
> 7. Principal component analysis on Grassmann manifolds [A7].
>
> **We agree that global geodesic convexity does not hold on positively curved manifolds such as the sphere. However, even in such cases, geodesically convex objectives arise on geodesically convex subsets.** For instance, squared geodesic distance functions are geodesically convex on geodesic balls of radius $<\pi/2$.
>
>
>
> >Q2. Regarding Line 3 of Algorithm 1, uniform random sampling on $S_{T_{x_k}M}(1)$
>  can become complex when applied to other manifolds, such as constant-rank manifolds. Is there a simpler method to replace this step?
>
>  We agree that tangent space projection is more computationally expensive for constant-rank manifolds compared to simpler cases such as spheres or SPD manifolds. Nevertheless, efficient formulas for these projections are well established [A2], and once available, sampling from $S_{T_X\mathcal M}(1)$ via normalized projected Gaussian is straightforward. Indeed, for $g \sim \mathcal N(0,I)$, the projected vector $\xi = \Pi_{T_x\mathcal M} g$ is $\mathcal N(0,I_{T_x\mathcal M})$ and thus rotationally invariant, implying $\xi/\|\xi\| \sim \mathrm{Unif}(S_{T_x\mathcal M}(1))$. **Thus, the main computational cost, arises from the tangent space projection rather than the sampling step itself.**
>
> **We also note that such costs arise even in first-order and zeroth-order methods [Li et al., 2023] and are difficult to avoid in our setting with weaker (dueling) feedback.** Importantly, this is not specific to our algorithm but is inherent to Riemannian optimization where tangent projections are routinely required for gradient computation, retractions, and vector transport, and thus form a standard computational bottleneck [A1,A2,Boumal, 2023]. In practice, for constant rank manifolds, this cost is often mitigated using factorized representations and randomized/approximate SVD schemes [A2,A3], which can be directly incorporated in our setting. **Consequently, our sampling step does not introduce additional overhead (up to constants) beyond what is already required by existing methods.**
>
>
> >Q3. Regarding the application scenarios for the RDNGD method, I am not entirely clear on the definition of a 'constrained convex problem where $f$ is geodesically convex and geodesically L-smooth.' Are there any concrete application examples of this in machine learning?
>
> **There are several concrete and widely studied applications in machine learning that can be formulated as constrained problems with a geodesically convex and geodesically $L$-smooth objective e.g.:**
>
> 1. Karcher Mean problem on SPD manifolds and Hyperbolic spaces [A7, A8].
>
> 2. Operator Scaling on SPD matrices [A7, A8].
>
> 3. Principal component analysis (PCA) on the Grassmannian manifold [A7].
>
> 4. Rayleigh Quotient problem with bounded constraints [A7].
>
>
> [A1] Absil, Mahony, and Sepulchre. Optimization algorithms on matrix manifolds. Princeton University Press, 2008.
>
> [A2] Vandereycken. "Low-rank matrix completion by Riemannian optimization." SIAM Journal on Optimization, 2013.
>
> [A3] Halko, Martinsson, and Tropp. "Finding structure with randomness: Probabilistic algorithms for constructing approximate matrix decompositions." SIAM review, 2011.
>
> [A4] Vishnoi. “Geodesic Convex Optimization: Differentiation on Manifolds, Geodesics, and Convexity.” arXiv:1806.06373 (2018).
>
> [A5] Vishnoi, and Yildiz. "On geodesically convex formulations for the brascamp-lieb constant." arXiv:1804.04051 (2018).
>
> [A6] Duembgen, and Tyler. "Geodesic convexity and regularized scatter estimators." arXiv:1607.05455 (2016).
>
> [A7] Wang, Tu, Hong, Wu, and Shi. “Online Optimization over Riemannian Manifolds.” JMLR, 2023.
>
> [A8] Kim, and Yang. "Accelerated gradient methods for geodesically convex optimization: Tractable algorithms and convergence analysis." ICML, 2022.

---

> > ### Author Rebuttal · Reviewer_AM8Q · 2026-04-03
> >
> > The authors provided many examples that hold under the current assumptions, and I am satisfied with their response.

---

> > > ### Author Response · Authors · 2026-04-03
> > >
> > > Thank you for acknowledging our clarifications and updating your score. We truly appreciate your time and constructive feedback.

---

### Official Review · Reviewer_VPXg · 2026-03-07

**Soundness:** 2
**Presentation:** 2
**Significance:** 2
**Originality:** 2
**Overall Recommendation:** 3
**Confidence:** 4

**Summary:**

This paper studied the dueling optimization over Riemannian manifolds, and proposed a class of Riemannian Dueling Normalized Gradient Descent (RDNGD) methods to solve these dueling optimization problems. It studied convergence properties of the proposed RDNGD methods. It also provided some numerical experiments to demonstrate effectiveness of the proposed methods.

**Compliance With Llm Reviewing Policy:**

Affirmed.

**Final Justification:**

My main concerns are basically solved.

However, the additional 8 pages of anonymous PDF (https://anonymous.4open.science/r/Additional_Experiments-B225/Additional_experiments.pdf) may be unfair to the authors of other articles.

So I keep my score.

**Key Questions For Authors:**

1)	In the abstract, the authors mention “we study dueling optimization over Riemannian manifolds, which covers important applications that cannot be solved by existing dueling optimization algorithms.” This expression is inaccurate. The existing dueling optimization algorithms should not easily or effectively solve these dueling optimization problems over Riemannian manifolds. In fact, the dueling optimization algorithms with some projected operators can solve these problems.

2)	This paper basically extends the existing Euclidean dueling optimization methods to the Riemannian manifold setting.  The authors should point novelties of the proposed methods.

3)	In the numerical experiments, many comparisons are missing. E.g., in the attacks on deep neural networks, some typical zeroth-order algorithms (such as ZO-ADMM) and the existing Euclidean Dueling Optimization methods are missing.

**Limitations:**

Yes

**Strengths And Weaknesses:**

Strengths:

1)	This paper presented a class of Riemannian Dueling Normalized Gradient Descent (RDNGD) methods to solve the dueling optimization problems over Riemannian manifolds.

2)	It studied convergence properties of the proposed methods under the geodesically L-smooth or geodesically (strongly) convex settings.

Weaknesses:

1)	This paper basically extends the existing Euclidean dueling optimization methods to Riemannian manifold setting. Meanwhile,  the  convergence analysis also follows the existing manifold optimization methods. Thus, the authors should point novelties of the proposed methods.

2)	In the numerical experiments, many comparisons are missing. E.g., in the attacks on deep neural networks, some typical zeroth-order algorithms such as ZO-ADMM and the existing Euclidean dueling optimization methods are missing.

---

> ### Author Rebuttal · Authors · 2026-03-31
>
> We thank the reviewer for the insightful questions, comments, and suggestions.
> > Q1. The authors mention “we study dueling optimization over Riemannian manifolds, which covers important applications that cannot be solved by existing dueling optimization algorithms.” ...the dueling optimization algorithms with some projected operators can solve these problems.
>
> We will clarify in the abstract that our intent is to convey that these problems are not solved efficiently or in a geometry-aware manner using Euclidean methods with projection as we explain below.
>
> **The reviewer suggests applying Euclidean dueling optimization with projection to Riemannian problems. This treats Riemannian optimization as a constrained Euclidean problem with a nonconvex constraint set (manifolds are nonconvex smooth surface), for which projection-based methods (e.g., [V3]) lack theoretical guarantees.** In contrast, our approach is grounded in Riemannian optimization theory and admits formal guarantees.
>
> **Moreover, projection would require solving a subproblem over a nonconvex set, which is NP-hard in general.** Even when heuristic projections exist (e.g., SPD manifolds), they are often computationally costly, requiring eigen decompositions or iterative solvers (Boumal, 2023; Zhang \& Sra, 2016). Moreover, **many objectives are nonconvex in the Euclidean sense but geodesically convex**, e.g., the Karcher mean on SPD. Projection ensures feasibility but does not overcome nonconvexity, whereas exploiting geodesic convexity **enables global guarantees unavailable in Euclidean formulations** (Zhang \& Sra, 2016).
>
> >Q2. This paper basically extends the existing Euclidean dueling optimization methods to the Riemannian manifold setting. The authors should point novelties of the proposed methods.
>
> We explain below why our contributions go beyond a simple direct extension in several important ways:
>
> 1. **Euclidean methods with projection are not equivalent to Riemannian methods.** As noted earlier, extending Euclidean dueling methods via projection would entail solving nonconvex projection subproblems, for which existing works offer no theoretical guarantee as they assume bounded convex constraint set [V3]. Moreover, such projection-based updates fail to respect intrinsic geometric structure (e.g., geodesic convexity). Riemannian methods are therefore fundamentally different and, in general, not reducible to Euclidean projection approaches.
> 2. **Computational advantages over projection-based approaches.** Prior work (Boumal, 2023; Zhang \& Sra, 2016) shows that Euclidean projection can be computationally prohibitive on manifolds (e.g., eigenvalue decompositions or iterative solvers for SPD/Stiefel). Riemannian methods avoid this entirely via intrinsic updates (e.g., RDNGD), eliminating the need for manifold projection. For constraints defined on the manifold, we propose RDFW, which uses linear minimization oracles that are more tractable than projection in certain cases (Weber & Sra, 2023). When specialized to Euclidean settings, RDFW yields a projection-free dueling-feedback method for constrained optimization, filling a gap in the Euclidean literature.
> 3. **Defining updates and estimators is intrinsically different.** In Riemannian settings, even defining the update and gradient estimator is non-trivial: linear perturbations $x\pm\nu u$ must be replaced by exponential maps, and their analysis requires curvature-dependent control via geodesic smoothness and parallel transport.
>
> 4. **Optimization dynamics depend on curvature.** Updates evolve along geodesics, and convergence depends explicitly on curvature (e.g., $\bar{\zeta}$), rather than only Lipschitz constants as in Euclidean settings.
>
> 5. **Improved guarantees even in the Euclidean limit.** Even when specialized to the Euclidean setting, our analysis yields improved constants and removes logarithmic factors compared to prior dueling optimization work.
>
>
> >Q3. In the numerical experiments, many comparisons are missing. E.g., in the attacks on deep neural networks, some typical zeroth-order algorithms (such as ZO-ADMM) and the existing Euclidean Dueling Optimization methods are missing.
>
> **We have conducted the suggested experiments. Please see our reply to Q4 of Reviewer 6LKY.**
>
> [V1] Bini, and Iannazzo. "Computing the Karcher mean of symmetric positive definite matrices." Linear Algebra and its Applications, 2013.
>
> [V2]Vishnoi. "Geodesic convex optimization: Differentiation on manifolds, geodesics, and convexity." arXiv:1806.06373 (2018).
>
> [V3] Saha, Koren, and Mansour. "Dueling Convex Optimization with General Preferences." ICML, 2025.
>
> [V4] Duembgen, and Tyler. "Geodesic convexity and regularized scatter estimators." arXiv:1607.05455 (2016).

---

> > ### Author Rebuttal · Reviewer_VPXg · 2026-04-02
> >
> > Thanks for your responses.
> >
> > You basically deal with my main concerns.
> >
> > Note that you have provided an additional 8 pages of anonymous PDF in rebuttal, **it likely violated the "5000-character limit" rule**.
> >
> > This additional 8 pages of anonymous PDF (https://anonymous.4open.science/r/Additional_Experiments-B225/Additional_experiments.pdf) **is unfair to the authors of other articles**.

---

> > > ### Author Response · Authors · 2026-04-02
> > >
> > > Thank you. We are glad that we could address your main concerns. To ensure a fair and constructive evaluation, we would appreciate it if you could specify any remaining concerns that prevent a full endorsement of the work. We are happy to provide further clarification if needed. If no concrete issues remain, we trust your final assessment will reflect that your concerns have been resolved.
> > >
> > > Regarding the additional anonymous PDF: this was provided specifically in response to requests from multiple reviewers (including yourself) for comparisons with Euclidean baselines and additional experiments. Providing an external anonymous link is permitted under ICML guidelines to present materials in rebuttal. Our intent was not to gain any unfair advantage, but rather to respond thoroughly and transparently to the reviewers’ requests. We hope this clarifies the motivation and context for including the link.

---

### Official Review · Reviewer_b6es · 2026-03-12

**Soundness:** 4
**Presentation:** 4
**Significance:** 3
**Originality:** 3
**Overall Recommendation:** 6
**Confidence:** 4

**Summary:**

This paper extends the duelling optimization scheme proposed by (Saha et al, 2021) to the Riemannian setting  while providing solid theoretical results analyzing iteration and oracle complexities for suitable classes of objective functions. As a byproduct of this extension, the specialization to the Euclidean case also provides a more refined result improving the results of (Saha et al, 2021). In cases where the projection step, that is typically a part of the proposed method, is computationally expensive, an alternative projection-free Frank-Wolfe type method is proposed along with its iteration and oracle complexities. The effectiveness of the proposed optimization methods is demonstrated on synthetic experiments along with a real world application.

**Compliance With Llm Reviewing Policy:**

Affirmed.

**Final Justification:**

As noted in my rebuttal acknowledgement, the authors have addressed the concerns raised in my initial review, and I do not have any remaining major issues with the paper. Overall, I find the work to be sound and clearly presented, and the clarifications provided in the rebuttal confirmed my opinion. I will therefore maintain my updated score of 6 as my final recommendation.

**Key Questions For Authors:**

I have the following minor clarifying questions or suggestions:

1. If I understood correctly, Lemma 3.1 gives a non-vacuous  statement only when $\sqrt{\frac{d}{2\pi}}L \nu \leq ||\textnormal{grad} f(x)||$.  I am wondering how does this fit in with the proof especially around the equilibrium when $||\textnormal{grad}  f(x)||$ is small?
2. Adding a comment on how the constants such (as the step size) in the numerical experiments are chosen in relation to the theory would be very helpful. This would connect the theoretical result more clearly with the numerical results. Even if not, clearly stating this in a Remark would be quite helpful.
3. Since $f$ is defined on $\mathcal{X}$ and not on $\mathcal{M}$, $x,y$ should be restricted to $ \mathcal{X}$ in equation (2)
4. The definition of sectional curvature can be included in the prelimiaries on Riemannian geometry in Appendix A
5. The footnote on page 4 spills into page 5. Perhaps this can be fixed such that it stays on page 4
6. Prohibited in the abstract should be prohibitive

**Limitations:**

As already discussed in weakness and key questions above, the connection between the theoretical resutls and empirical results can be spelled out more clearly, especially in terms of the choices of the parameters such as the step size. Other than that, a remark on how strict the present assumptions on the function class are and if/how these can be further relaxed would also be valuable.

**Strengths And Weaknesses:**

### Strengths
1. The paper is well written, easy to follow and addresses an important class of optimization problems relevant for the readership of ICML.
2. The proposed optimization method is backed up by formal theoretical results as well empirical studies supporting the claims very well. The assumptions made on the function class for the theoretical results are commonly found in the literature and are not overly restrictive.
3. Detailed proofs of the theoretical results are made available in the Appendix (although I must admit that I did not verify the proofs)

### Weakness:
The theoretical results specify a particular choice of step size (either constant with a prescribed value of cosine annealing) under which we get the desirable convergence results. The step sizes chosen in Section 5 seem unrelated to these choices prescribed by theory. In this sense, the theoretical results do not seem to inform the choice of step size. I suspect the reason for this is that the prescribed step size cannot be computed a priori due to the lack of knowledge of the constants involved. If this is the case, the theoretical results lose their practical utility. I see that they still contribute towards an understanding of the method. A desirable improvement could come in the form of theoretical results that prove the implications under the choice of step size within some interval,  say $[0,\eta_{\textnormal{max}}]$ with $\eta_{\textnormal{max}}$ being the presently prescribed value. In this case, even though one may not know $\eta_{\textnormal{max}}$ exactly, one can always choose a step size small enough for the theorem to be valid. I see that the original work (Saha et al, 2021) also has this problem but it leaves me wondering if the proof technique can be adapted to a larger class of step sizes even in this work.

---

> ### Author Rebuttal · Authors · 2026-03-30
>
> We thank the reviewer for the insightful questions, comments, and suggestions.
>
> >Weakness: "...prescribed step size cannot be computed a priori due to the lack of knowledge of the constants ...prove the implications under the choice of step size within some interval, say $[0,\eta_{max}]$ with $\eta_{max}$..."
>
> **Our results already establish exactly what is suggested: the algorithm converges for all step sizes $\eta \in (0,\eta_{\max}]$.** We only state $\eta_{\max}$ in the theorem because it yields the fastest convergence rate, but the proofs work for any smaller step size, with a corresponding slowdown in the rate. This is explicit in Eq.~(25) for the constant step-size case and lines 1000--1009 on page 19 for the cosine schedule. We will clarify this in the main theorem statement.
>
> **The issue still remains that $\eta_{\max}$ indeed depends on unknown problem parameters (e.g., smoothness). However, this is standard across optimization methods (see [B1] for a survey and the references therein).** One way to resolve this is to use parameter-agnostic methods (e.g., [B2]). These methods often incur additional costs such as exponential dependence on smoothness parameters. **Alternatively, one can use diminishing step sizes** with appropriately chosen decay rate to preserve the convergence rate. This ensures that, even if a step-size larger than $\eta_{\text{max}}$ is chosen at initialization, the step-sizes will eventually become smaller than $\eta_{\text{max}}$. However, practically this can be slow.
>
> **To illustrate that our proofs work with diminishing step sizes, we provide a version of part (i) of Theorem 3.7 using diminishing step sizes.** Due to the character limit, we will not include the proof here. In this version $\nu$ depends on unknown problem parameters but unlike step-size, $\nu$ can be chosen small without slowing down convergence. In practice, it just suffices to choose some small $\nu\sim 10^{-8}$.
>
> *Theorem 3.7(i) with diminishing step-sizes: Assume $f$ is geodesically $L$-smooth and geodesically convex, and Assumptions 2.5 and Assumption 2.6 hold. Consider using RDNGD to solve target problem over a bounded set $\mathcal{X}\subset\mathcal{M}$ of diameter $\mathfrak{D}$. For any $\epsilon>0$, RDNGD returns an $\epsilon$-optimal solution satisfying $ \mathbb E[f(\hat x_T)] - f(x^\*) < \epsilon $ with the following inputs.*
> \begin{equation}
> \begin{split}
>     T = O(\frac{1}{\epsilon}\log(\frac{1}{\epsilon})),
>     \eta_k=\frac{1}{(k+1)^{1/2}},
>     \nu = \frac{1}{4\sqrt{2} d}\frac{(\epsilon/{L})^{3/2}}{(\sqrt{D}+\eta T)^2}.
> \end{split}
> \end{equation}
>
> >Q1. Lemma 3.1 gives a non-vacuous statement only when $\sqrt{d/2\pi}L\nu\leq \||grad f(x)\||$ ... how does this fit in with the proof especially ... when $\||grad f(x)\||$ is small?
>
> We agree that Lemma 3.1, when viewed in isolation, can appear vacuous near stationarity. However, in our analysis the goal is to obtain an $\epsilon$-stationary point or $\epsilon$-approximate minima, i.e., $\||\mathrm{grad }f(x)\|| \leq \epsilon$. Accordingly, $\nu$ is chosen as a function of $\epsilon$ (see, e.g., Eq.~(24)), specifically $\nu <\frac{\sqrt{2\pi}\epsilon}{L\sqrt{d}}$. **Under this choice, we ensure that $\gamma_x < 1$ throughout the regime of interest, and in fact $\gamma_x$ can be made arbitrarily small by taking $\nu$ smaller. Therefore, the condition in Lemma 3.1 does not pose a limitation in the proof, including near equilibrium.** We will clarify the choice of $\nu$ in Lemma 3.1.
>
> >Q2. Adding a comment on how the constants such (as the step size) in the numerical experiments are chosen in relation to the theory would be very helpful.
>
> We will add a remark to address this part highlighting the following points:
> 1. **We used a standard grid search to select the step size.**
> 2. For cosine annealing, except for $\eta_0$  (chosen by grid search), other parameters (perturbation scale $\nu$ and $\eta_{\min}$ ) show negligible impact when set below $10^{-6}$ and $10^{-8}$, respectively.
> 3. For comparison with other methods, we followed their original problem setups, including dimensions, initializations, and all parameters. To ensure a fair comparison, all baselines were also tuned via a simple grid search around the recommended configurations in the respective original papers.
>
> >Q3. Since $f$ is defined on $\mathcal{X}$ ... oracle should be restricted to $\mathcal{X}$...
>
> We agree. Whenever $\mathcal{Q}_f(x,y)$ is used, $x,y \in \mathcal{X}$, so restricting the oracle to $\mathcal{X}\subset\mathcal{M}$ suffices. We will revise Equation (2) accordingly.
>
> >Q4-6. Minor changes.
>
> We will include the definition of sectional curvature and change the footnote and the wording accordingly.
>
> [B1] Guillaume, and Gower. "Handbook of convergence theorems for (stochastic) gradient methods." arXiv preprint arXiv:2301.11235 (2023).
>
> [B2] Hübler, Yang, Li, and He. "Parameter-agnostic optimization under relaxed smoothness." AISTATS, 2024.

---

> > ### Author Rebuttal · Reviewer_b6es · 2026-04-01
> >
> > Thank you for your thorough response and for addressing all of my concerns. I appreciate the clarifications provided.
> >
> > Regarding the suggestion to include additional details on the choice of step size and related constants: my original question was mainly aimed at understanding whether these choices were informed by the theoretical analysis. If, however, these parameters were selected via grid search, I do not consider it necessary to elaborate further on how the grid was constructed, although such details could still be informative for readers.
> >
> > As I do not have any remaining concerns or open questions, I will update my score to 6 while maintaining my confidence level at 4.

---

> > > ### Author Response · Authors · 2026-04-01
> > >
> > > Thank you so much for the acknowledgement and increasing our score. We really appreciate it.

---

### Official Review · Reviewer_6LKY · 2026-03-13

**Soundness:** 3
**Presentation:** 3
**Significance:** 3
**Originality:** 3
**Overall Recommendation:** 4
**Confidence:** 3

**Summary:**

This paper considers Dueling optimization (pairwise comparison oracle only) on Riemannian manifolds. The authors consider several algorithms for diffrent function classes defined on Riemannian manifolds and establish corresponding oracle complexities. Numerical experiments are performed on several tasks: Rayleigh quotient, Karcher mean, adversarial attacks, and horizon leveling.

**Compliance With Llm Reviewing Policy:**

Affirmed.

**Final Justification:**

This work pioneers the study of dueling optimization in the Riemannian setting. Given the paper’s combination of theoretical depth and empirical validation, it is well-suited for the ICML community. I vote for acceptance.

**Key Questions For Authors:**

- How would gradient estimator and bounds degrade with stochastic oracle (e.g., Bradley-Terry)?
- Can variance reduction or capping $M_k$ give constant batch size for RDFW?
- Do convex guarantees (Thm 3.7) apply to Rayleigh quotient and adversarial attack (sphere), or only non-convex rates?
- Comparison to naive Euclidean dueling (NGD in ambient space then project)?
- How does batching (e.g., batch 10 in CIFAR-10) affect oracle complexity and single-sample performance in high dimensions?

**Limitations:**

The paper should explicitly acknowledge the deterministic-oracle assumption and discuss how stochastic feedback (e.g., human preferences) could affect the methods. It should also note the scalability issue of RDFW due to the growing $M_k$, and the practical difficulty of tuning hyperparameters when geometric constants such as $L$, $\kappa$, and $D$ are unknown and function values are unavailable for line search.

**Strengths And Weaknesses:**

**Strengths:** This is the first work to study dueling optimization algorithms intrinsically on Riemannian manifolds. The theoretical analysis is rigorous and improves the gradient-alignment constant $\hat{C}$ over prior Euclidean results. The presentation is clear, Table 1 effectively summarizes the contributions, and the problem is well motivated by preference-based learning applications.

**Weaknesses:** The theory assumes a deterministic comparison oracle, whereas human feedback is inherently stochastic (e.g., Bradley-Terry models). RDFW requires a batch size $M_k$ that grows linearly with $k$, which can be prohibitively expensive when each query is costly. There is also a gap between the convex theory (for Hadamard manifolds) and experiments on spheres, and the paper lacks a naive Euclidean dueling baseline for comparison.

---

> ### Author Rebuttal · Authors · 2026-03-30
>
> We thank the reviewer for the insightful questions, comments, and suggestions.
> >Q1. How would gradient estimator and bounds degrade with stochastic oracle (e.g., Bradley-Terry)?
>
> We agree that Bradley–Terry (BT) models are widely used. However, **they provide richer information (gradient magnitude and direction both) than our model-free deterministic oracle, which avoids BT’s drawbacks [L6] but only recovers direction.** Indeed, under BT-type model, the preference probability between $x_1$ and $x_2$ is given by
> $$P(x_1 \succ x_2) =p_{x_1,x_2}= \sigma(f(x_1)-f(x_2)),$$  for some link function $\sigma$. Querying $(x_1,x_2)=(x,Exp_x(\nu u))$, where $u=Pu_0$ is the projection of $u_0\sim N(0,I)$ on $T_x\mathcal{M}$, one can estimate $f(Exp_x(\nu u))-f(x)$ by $\sigma^{-1}(\hat p_{x_1,x_2})$ where $\hat p_{x_1,x_2}$ is the estimate of $p_{x_1,x_2}$. Then zeroth-order estimator can be used to estimate gradient magnitude and direction both unlike our setting where only gradient direction can be estimated. A more interesting case would be stochastic model-free dueling feedback as considered in Saha et al., 2021. But extending that to manifold setting is nontrivial due to the stochasticity-curvature interaction and we leave that to future work.
> >Q2. Can variance reduction or capping  give constant batch size for RDFW?
>
> **The randomness in our estimator arises from perturbations used to estimate gradient direction from comparisons, not from data subsampling. Hence standard VR techniques for finite-sum settings that rely on unbiased gradients, gradient differences [L2, L4], or even second-order information [L5] are inapplicable as none of those structures is available.** Moreover, since we use gradient direction (unit norm) to update parameters, capping does not help.
>
>
> **Moving-average VR** is also ineffective. It **relies on smoothness to ensure $\nabla f(x_k)\approx \nabla f(x_{k+1})$ when iterates are close, but this logic does not apply to normalized gradients.** Moreover, since we only observe the direction
> $v=\nabla f(x)/\||\nabla f(x)\||_2$, not $\||\nabla f(x)\||_2$, we cannot do VR before normalization either.
>
> So we use increasing batch sizes. This is standard in stochastic FW methods [L2,L3,L4]. Designing VR for comparison-based estimators (e.g., using structured reuse of perturbations) is an interesting direction, but we leave it to future work as it appears substantially more challenging.
> >Q3. Do convex guarantees (Thm 3.7) apply to Rayleigh quotient and adversarial attack (sphere), or only non-convex rates?
>
> For Rayleigh quotient and adversarial attack examples, we obtain non-convex rates.
>
> >Q4. Comparison to naive Euclidean dueling (NGD in ambient space then project)?
>
> Expt_Link=https://anonymous.4open.science/r/Additional_Experiments-B225/Additional_experiments.pdf
>
> **Projected Dueling NGD (PDNGD), i.e., ambient dueling NGD followed by projection, requires projection onto nonconvex set, and thus lacks theoretical convergence guarantees** (see response to Q1 of Reviewer VPXg). So we compare empirically with PDNGD on the sphere (where projection is available) and on SPD manifolds, where projection is heuristically approximated (Section 1.2 in Expt_Link).
>
> **Rayleigh quotient maximization:** Figures 1-2. Across various cosine stepsizes, RDNGD consistently achieves higher final accuracy while PDNGD shows a marginal advantage in CPU time.
>
> **Karcher mean:** Figures 3-4. RDNGD-cosine yields better precision and efficiency than PDNGD-cosine across all configurations.
>
> **Adversarial attack:** Figures 8-10. We compare our proposed RDNGD with PDNGD and ZO-ADMM. RDNGD achieves similar precision and CPU time to PDNGD. While ZO-ADMM outperforms RDNGD in certain cases, RDNGD only requires a comparison oracle, whereas ZO-ADMM relies on explicit function values, and hence inapplicable when only comparison oracle is available.
> >Q5. How does batching (e.g., batch 10 in CIFAR-10) affect oracle complexity and single-sample performance in high dimensions?
>
> We conduct an ablation over batch sizes $m \in \lbrace 1, 5, 10, 20, 30, 50\rbrace$ (Figures 5–7 in Section 2.1 in Expt_Link). While $m=1$ succeeds in one case, it fails within the iteration budget in others. Larger $m$ reduces iterations but increases per-iteration cost; overall, $m\in \lbrace 5,10\rbrace$ achieves the best CPU time.
>
> [L1] Li, Balasubramanian, and Ma. "Stochastic zeroth-order Riemannian derivative estimation and optimization." MOR, 2023.
>
> [L2] Hazan, and Luo. “Variance-Reduced and Projection-Free Stochastic Optimization.” ICML, 2016.
>
> [L3]Reddi, Sra, Póczos, and Smola. “Stochastic Frank–Wolfe Methods for Nonconvex Optimization.” Allerton CCCC, 2016.
>
> [L4] Goldfarb, Iyengar, and Zhou. "Linear convergence of stochastic frank wolfe variants." AISTATS, 2017.
>
> [L5] Zhang, et.al. "One sample stochastic frank-wolfe." AISTATS, 2020.
>
> [L6] Zhang, e.al. "Beyond bradley-terry models: A general preference model for language model alignment." ICML, 2025.

---

> > ### Author Rebuttal · Reviewer_6LKY · 2026-04-04
> >
> > Thank you for your detailed response. I will keep my score.

---

> > > ### Author Response · Authors · 2026-04-05
> > >
> > > Thank you. We appreciate it. If there are any remaining aspects we could further clarify or improve to strengthen your assessment, we would be glad to address them.

---

### Decision · Program_Chairs · 2026-04-30

**Decision:**

Accept (regular)

**Comment:**

This paper applies dueling optimization theory to the domain of Riemannian manifolds, effectively addressing the computational and theoretical limitations in standard Euclidean projection methods. The reviewers acknowledged the work's solid theoretical foundations, as well as the effective experiments conducted across various tasks. During the rebuttal phase, the authors provided comparative experiments against Euclidean baseline methods (such as PDNGD and ZO-ADMM), further clarified the rationale behind their step-size selection, and elucidated the specific application scenarios for geodesic convexity, thereby substantially resolving the reviewers' primary concerns. In conclusion, given the paper's significant theoretical novelty and practical value, I recommend its acceptance as a poster presentation.